# UV-Attack: Physical-World Adversarial Attacks on Person Detection via Dynamic-NeRF-based UV Mapping

**Yanjie Li, Kaisheng Liang, Bin Xiao** *
Department of Computer Science, Hong Kong Polytechnic University
`yanjie.li@connect.polyu.hk`
`cskliang@comp.polyu.edu.hk`
`b.xiao@polyu.edu.hk`

## Abstract

In recent research, adversarial attacks on person detectors using patches or static 3D model-based texture modifications have struggled with low success rates due to the flexible nature of human movement. Modeling the 3D deformations caused by various actions has been a major challenge. Fortunately, advancements in Neural Radiance Fields (NeRF) for dynamic human modeling offer new possibilities. In this paper, we introduce `UV-Attack`, a groundbreaking approach that achieves high success rates even with extensive and unseen human actions. We address the challenge above by leveraging dynamic-NeRF-based UV mapping. `UV-Attack` can generate human images across diverse actions and viewpoints, and even create novel actions by sampling from the SMPL parameter space. While dynamic NeRF models are capable of modeling human bodies, modifying clothing textures is challenging because they are embedded in neural network parameters. To tackle this, `UV-Attack` generates UV maps instead of RGB images and modifies the texture stacks. This approach enables real-time texture edits and makes the attack more practical. We also propose a novel Expectation over Pose Transformation loss (EoPT) to improve the evasion success rate on unseen poses and views. Our experiments show that `UV-Attack` achieves a 92.7% attack success rate against the FastRCNN model across varied poses in dynamic video settings, significantly outperforming the state-of-the-art AdvCamou attack, which only had a 28.5% ASR. Moreover, we achieve 49.5% ASR on the latest YOLOv8 detector in black-box settings. This work highlights the potential of dynamic NeRF-based UV mapping for creating more effective adversarial attacks on person detectors, addressing key challenges in modeling human movement and texture modification. The code is available at https://github.com/PolyLiYJ/UV-Attack.

## 1 Introduction

Adversarial attacks have become significant concerns in both digital and physical domains, impacting areas like facial recognition (Gong et al., 2023; Yang et al., 2023) and object detection (Huang et al., 2023; Li et al., 2023c). To improve the physical attack success rate, researchers have attempted to use 2D transformations-such as translation, rescaling, and shearing (Zhong et al., 2022)-and 3D modeling (Huang et al., 2024; Suryanto et al., 2022) to simulate changes in viewpoints for objects like stop signs and cars. However, compared with attacks on rigid objects, attacks on non-rigid objects like human beings are more challenging because of the variability in clothing distortions and human poses. These non-rigid distortions are difficult to simulate using basic transformations or static 3D modeling. Consequently, previous attempts at person detection attacks often relied on adding an adversarial patch to the front of the T-shirt (Xu et al., 2020; Wang et al., 2021b; Lin et al., 2023) or assumed minimal movement (Hu et al., 2022; 2023), which limits the real-world attack success rate when the subject undergoes significant movement and unseen poses.

---

*Corresponding Author

To address these challenges, in this work, we propose the `UV-Attack`, which successfully addresses this challenge by incorporating dynamic NeRF-based UV mapping into the attack pipeline. Recent advancements have applied neural radiance fields (NeRF) (Barron et al., 2021) to dynamically modeling human bodies (Geng et al., 2023; Weng et al., 2022). These methods enable the modeling of human bodies as neural networks and generate novel poses with minimal parameters. Nevertheless, directly modifying the texture of NeRF models is challenging due to the texture information being embedded within neural network parameters (Dong et al., 2022). To solve this issue, inspired by the latest work of editable NeRF model (Chen et al., 2023c), we model the 3D human body's shape and textures respectively. Then we focus on editing the 3D textures through learnable Densepose-like UV-maps (Güler et al., 2018). This approach enables real-time editing of 3D textures without requiring gradient backpropagation on the NeRF model. Moreover, to improve the physical attack success rate on *unseen* poses, we sample novel poses from the SMPL parameter spaces and generate human images with unobserved poses and viewpoints. Moreover, we propose a novel Expectation over Pose Transformation(EoPT) loss function to enhance the attack's effectiveness.

Furthermore, recent studies have explored the use of diffusion models to enhance the transferability of adversarial examples (Xue et al., 2023; Lin et al., 2023; Chen et al., 2023b). However, these methods show limited success in physical attacks due to restricted noise levels. To address this problem, we improve the physical attack success rate by expanding the perturbation space. By tweaking the starting point of the stable diffusion model and removing classifier-free guidance, we achieve a higher ASR, as our method allows for the exploration of larger and less familiar areas in the latent space compared to previous approaches.

We compare `UV-Attack` with the latest person detection attacks (Hu et al., 2022; 2023) and diffusion-based attacks (Xue et al., 2023; Lin et al., 2023). Our experiments demonstrate the superiority of our approach in terms of evasion success rates in free-pose scenarios. As shown in Figure 1, we physically printed the Bohemian-style clothing and successfully fooled person detectors in dynamic video settings, even when the subjects were in continuous and significant movement. Our key contributions are:

1. We propose a novel person detection attack algorithm that effectively handles pose variation through UV mapping. Our method can generate unseen actions and can edit human clothes in real time. Moreover, we propose a novel Expectation over Pose Transformation loss (EoPT) to improve the evasion success rate on unseen poses and views.

2. We utilize the stable diffusion model to enhance the transferability of adversarial patches. We expand the search space by adjusting the starting point of the conditional latent diffusion model without employing classifier-free guidance, which results in a higher physical ASR.

3. Our attack is designed to be physically realizable. We manufacture adversarial clothing in the real world and achieve a physical ASR of 85% in a free-pose setting. This significantly surpasses the performance of the state-of-the-art Adv-Camou (Hu et al., 2023) attack, which attains only a 25% ASR. Moreover, we achieve 49.5% ASR on the latest YOLOv8 detector in black-box settings.

## 2 BACKGROUND

**Adversarial Attacks against Object Detection** Recent work has employed elaborate adversarial examples to attack real-world image classifiers (Wang et al., 2023; Zhong et al., 2022) and object detectors (Zhang et al., 2023; Huang et al., 2023; Li et al., 2023c; Wang et al., 2022). Unlike digital attacks, physical attacks introduce greater complexity due to the physical variables. Early efforts to improve the efficacy using basic 2D transformations like translation, rescaling, and shearing (Zhao et al., 2019; Zhong et al., 2022). However, these approaches are largely effective only for planar objects such as stop signs. For 3D objects, more advanced techniques involving static 3D modeling and differentiable rendering have been employed to modify the textures (Wang et al., 2022; 2021a; Ravi et al., 2020). However, such methods are generally limited to rigid objects and fail to adequately address the dynamic and non-rigid targets. Specifically, static 3D models do not effectively simulate human poses, and the use of 2D and 3D Thin Plate Splines (TPS) Xu et al. (2020) has only been successful in modeling minor clothing distortions, falling short in replicating the impact of diverse human poses. Consequently, most previous person detection attacks have merely added an adversarial patch to the front of the T-shirt Hu et al. (2021); Duan et al. (2022); Tan et al. (2021) to avoid distortion

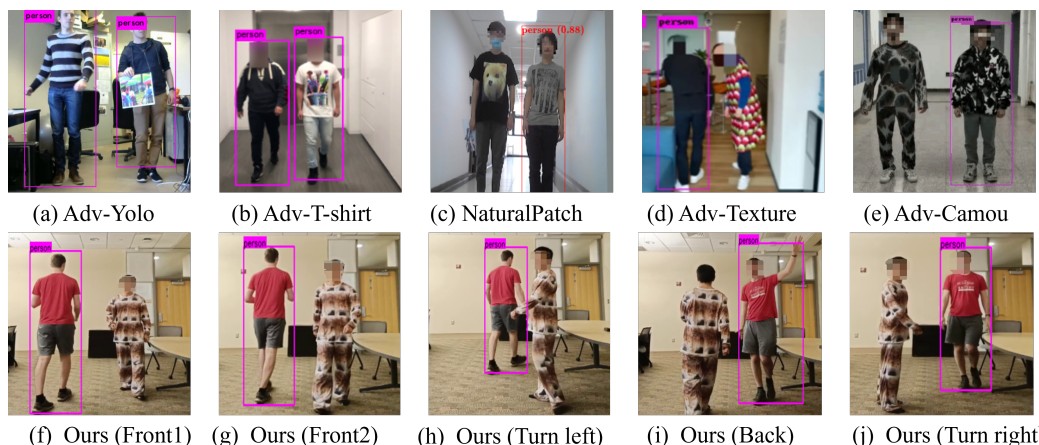

(a) Adv-Yolo    (b) Adv-T-shirt    (c) NaturalPatch    (d) Adv-Texture    (e) Adv-Camou

(f) Ours (Front1)    (g) Ours (Front2)    (h) Ours (Turn left)    (i) Ours (Back)    (j) Ours (Turn right)

Figure 1: Visualization of different person detector attacks. We compare our attack with different person detection attacks, including Adv-Yolo (Thys et al., 2019), Adv-Tshirt (Xu et al., 2020), NaturalPatch (Hu et al., 2021), Adv-Texture (Hu et al., 2022) and Adv-Camou (Hu et al., 2023). Our attacks achieve higher ASR under diverse poses and viewpoints than previous attacks.

problems. More recent approaches have incorporated 3D texture-based modifications Hu et al. (2023; 2022) to improve attack success rates from varying viewpoints. However, these approaches relied on static modeling and did not consider the influence of human poses. This highlights the critical challenge of creating novel attack techniques that effectively address the dynamic nature of human movements and pose variations.

**Adversarial Attacks based on NeRF** Neural Radiance Fields (NeRF) (Mildenhall et al., 2021) have demonstrated significant potential in rendering free-view images, making them attractive for physical attacks. Recent methodologies have explored the integration of NeRF in various adversarial contexts. ViewFool (Dong et al., 2022) leverages NeRF to find adversarial viewpoints that can generate adversarial images. It does not modify the object itself. VIAT (Ruan et al., 2023) follows ViewFool by learning a Gaussian mixture distribution of adversarial viewpoints to cover multiple local maxima of the loss landscape. Adv3D (Li et al., 2023a) utilizes Lift3D (Li et al., 2023b) to alter the texture latent of NeRF models, specifically targeting the textures of cars. However, it only considers rigid objects and is not physically realizable. TT3D (Huang et al., 2024) also considers the rigid objects. It modifies texture latent within grid-based NeRF architectures (Müller et al., 2022). Their method can change the textures but cannot be applied to non-rigid objects. In contrast to previous methods, our attack is the first specifically designed to target person detection. Unlike attacks focused on rigid objects, modeling a person is more challenging due to non-rigid deformations. `UV-Attack` is the first to utilize dynamic NeRF and UV mapping in non-rigid 3D adversarial attacks.

## 3 METHODOLOGY

In this section, we detail the proposed `UV-Attack`. `UV-Attack` adopts the dynamic NeRF to model humans as digital representations and perform the optimization in the latent space of the diffusion model to generate the adversarial textures. In the following, we first introduce the background knowledge of dynamic NeRF and then present our methodology.

### 3.1 PRELIMINARY: DYNAMIC NEURAL RADIANCE FIELDS

NeRF enables novel-view photorealistic synthesis by modeling 3D objects as a continuous volumetric radiance field, allowing control over illuminations and materials (Mildenhall et al., 2021). It can be formulated as $F : (\mathbf{x}, \mathbf{d}) \rightarrow (\mathbf{c}, \sigma)$, where $F$ is a MLP and takes the 3D coordinate $\mathbf{x} \in \mathbb{R}^3$ and the view direction $\mathbf{d} \in \mathbb{R}^3$ as input and outputs the corresponding color $\mathbf{c}$ and volume density $\sigma$. Then NeRF computes the pixel color $C(\mathbf{r})$ by accumulating all colors along a ray $\mathbf{r} = \mathbf{o} + t\mathbf{d}$ that emitting

from the camera's original point $o$ and passing the pixel with near and far bounds $t_n$ and $t_f$,

$$C(\mathbf{r}) = \int_{t_n}^{t_f} T(t) \cdot \sigma(\mathbf{r}(t)) \cdot c(\mathbf{r}(t), \mathbf{d}) \, dt, \ \text{ where } T(t) = \exp\left(-\int_{t_n}^{t} \sigma(\mathbf{r}(s)) \, ds\right), \quad (1)$$

where $\sigma$ is the volume density at any point $s$ along the ray. $\mathbf{c}(\mathbf{r}(t), \mathbf{d})$ and $\sigma(\mathbf{r}(t))$ are the color and density for the 3D coordinate $\mathbf{r}(t)$ and view direction $\mathbf{d}$.

More recently, dynamic NeRF models have been proposed to accurately depict non-rigid objects like human beings (Geng et al., 2023; Weng et al., 2022). In response to the need for fast rendering and editable textures, we utilize UV-Volume (Chen et al., 2023c) to model the human body.

The core idea of UV-Volume is to render a NeRF model into UV maps rather than RGB images. UV maps contain UV coordinates (or texture coordinates), which are assigned pixel coordinates of a predefined texture image $\mathcal{E}$. The UV mapping process assigns pixels in the predefined texture image $\mathcal{E}$ to a rendered RGB image according to the UV coordinates. The UV-Volume input is the pose parameter $\theta$, the camera position $\eta$ and the light parameter $\tau$. It firstly computes a pixel feature $\mathcal{F}(\mathbf{r})$ by a rendering equation similar to Eq. 1, where $\mathbf{r}$ is a camera ray. Then $\mathcal{F}(\mathbf{r})$ is fed into a UV decoder and is decoded as part assignment and UV coordinates through $[\hat{\mathcal{P}}(\mathbf{r}), \hat{\mathcal{U}}(\mathbf{r}), \hat{\mathcal{V}}(\mathbf{r})] = M_{uv}(\mathcal{F}(\mathbf{r}))$. $\hat{\mathcal{P}}$ is a part assignment corresponding to different human body parts.

UV-volume also incorporates a texture generator, denoted by $G$, to generate texture stack images $\mathcal{E}_k = G(\theta, \boldsymbol{k})$, where $\boldsymbol{k}$ is a one-hot vector that specifies the body part (e.g., arm, leg). A texture feature, $\mathbf{e}_k$, is then sampled from $\mathcal{E}_k$ based on the UV coordinate using the differentiable grid sampling: $\mathbf{e}_k(\mathbf{r}) = \mathcal{E}_k[\hat{\mathcal{U}}_k(\mathbf{r}), \hat{\mathcal{V}}_k(\mathbf{r})]$. Finally, the RGB color for the camera ray $\boldsymbol{r}$ is calculated as a weighted sum of individual body part colors $\hat{C}_k(\mathbf{r})$: $\hat{C}(\mathbf{r}) = \Sigma_{k=1}^{K} \hat{\mathcal{P}}_k(\mathbf{r})\hat{C}_k(\mathbf{r})$. Each $\hat{C}_k(\mathbf{r})$ is determined by an MLP, $M_c$, which takes as input the UV coordinates (with positional encoding), the sampled texture feature, and the encoded view direction:

$$\hat{C}_k(\mathbf{r}) = M_c\left(\gamma(\hat{\mathcal{U}}_k(\boldsymbol{r}), \hat{\mathcal{V}}_k(\boldsymbol{r})), \mathbf{e}_k(\mathbf{r}), k, \gamma(\boldsymbol{d})\right), \quad (2)$$

where $K$ is the total number of human body parts and $\gamma(\cdot)$ a positional encoding function. The $M_c$ is an MLP that maps UV coordinates, sampled texture pixels and view directions into RGB images.

## 3.2 THE PIPELINE OF UV-ATTACK

We present the pipeline of UV-Attack in Figure 2. UV-Attack first samples poses, camera, and lighting parameters to generate IUV maps and texture stacks. Then it creates an adversarial patch using a pre-trained Stable Diffusion model and uses it to modify the texture stacks. These textures are then applied to the IUV maps, resulting in a final adversarial image.

**GMM-based Pose Sampling**   To generate unseen poses for the target person, we use a Gaussian Mixture Model (GMM) to fit a human pose distribution on target person images and an annotated pose dataset MPII (Andriluka et al., 2014). Then we sample pose parameters from this Gaussian Mixture distribution as the UV-Volume model's input. Specifically, We first use the SPIN model (Kolotouros et al., 2019) to estimate the poses $\hat{\boldsymbol{\theta}}$ and shapes parameters $\hat{\boldsymbol{\beta}}$ from the target person's video and add auxiliary poses $\tilde{\boldsymbol{\theta}}$ from public annotated poses datasets like MPII (Andriluka et al., 2014). Then we parameterize the distribution $p([\hat{\boldsymbol{\theta}}, \tilde{\boldsymbol{\theta}}])$ by a mixture of $K$ Gaussian components and take the transformation of random variable approach to ensure the support of $p([\hat{\boldsymbol{\theta}}, \tilde{\boldsymbol{\theta}}])$ is in scope of $[\boldsymbol{\theta}_{min}, \boldsymbol{\theta}_{max}]$ by

$$\boldsymbol{\theta} = \boldsymbol{a} \cdot \tanh(\boldsymbol{u}) + \boldsymbol{b}, \ \text{ where } \ p(\boldsymbol{u} \mid \boldsymbol{\Psi}) = \sum_{k=1}^{K} \omega_k \mathcal{N}(\boldsymbol{u} \mid \boldsymbol{\mu}_k, \boldsymbol{\sigma}_k^2 \boldsymbol{I}), \quad (3)$$

where $\boldsymbol{a} = \frac{(\boldsymbol{\theta}_{\max} - \boldsymbol{\theta}_{\min})}{2}$, $\boldsymbol{b} = \frac{(\boldsymbol{\theta}_{\max} + \boldsymbol{\theta}_{\min})}{2}$, $\boldsymbol{\Psi} = \{\omega_k, \boldsymbol{\mu}_k, \boldsymbol{\sigma}_k\}_{k=1}^{K}$ are the parameters of the GMM with weight $\omega_k \in [0, 1] \left(\sum_{k=1}^{K} \omega_k = 1\right)$. The $\boldsymbol{\mu}_k$ and $\boldsymbol{\sigma}_k$ are parameters of $k$-th Gaussian component.

**Real-Time Dynamic Texture Modification**   To improve the speed of adversarial attacks, it's crucial to efficiently modify and render clothing on 3D human models. We slightly adjust the UV-Volume

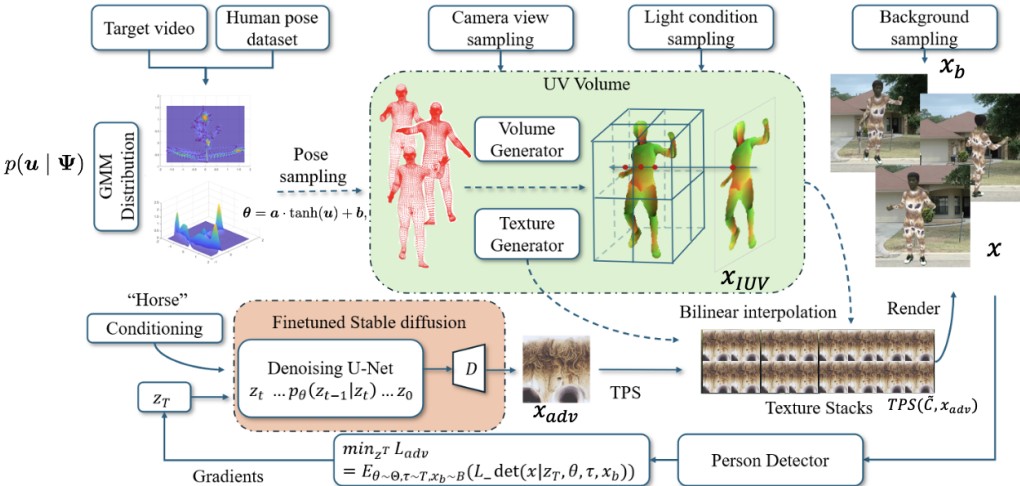

Figure 2: The pipeline of UV-Attack. UV-Attack first samples random pose parameters from a Gaussian Mixture Model. We then use the sampled pose, camera, and light parameters to generate IUV maps and texture stacks. We employ a pre-trained stable diffusion model and modify the initial latent to generate adversarial patches. Finally, we modify the texture stacks using the adversarial patch and get the human image in RGB space. The solid lines below indicate the direction of gradient propagation, while the dashed lines above do not require gradients.

architecture to further improve the texture modification speed. Specifically. we firstly define linearly spaced vectors $\mathbf{u}, \mathbf{v} \in \text{linespace}(0, 1, N_t)$, where $N_t$ is the texture's size. Then we create 2D grids using a mesh grid defined as $[\tilde{\mathcal{U}}, \tilde{\mathcal{V}}] = [\mathbf{1}_{N_t} \otimes \mathbf{u}, (\mathbf{1}_{N_t} \otimes \mathbf{v})^T]$, where is $\mathbf{1}_{N_t}$ is a column vector of ones with dimension $N_t$ and $\otimes$ denotes the Kronecker product. Then we remove the viewing direction $d$ from Eq.2 to generate the view-independent texture stacks Besides the texture stack $\tilde{C}$, we also generate IUV maps according to the pose, camera, and light parameters, which can be expressed $x_{\text{IUV}} = [\hat{\mathcal{P}}, \hat{\mathcal{U}}, \hat{\mathcal{V}}] = M_{uv}(\mathcal{F}_{\theta, \eta, \tau})$, where $\theta$ is the pose parameter and $\eta, \tau$ are the camera and light parameters.

During the process of image rendering for new actions, we randomly sample the camera views from different elevation and azimuth angles and different light positions and backgrounds. The rendered images can be expressed as

$$x = g(x_{\text{IUV}}(\theta, \tau), \text{TPS}(\tilde{C}, x_{adv})) \odot M + x_{\text{b}} \odot (1 - M), \quad (4)$$

where $x_{adv}$ is an adversarial patch generated by the latent diffusion model, $\tau$ is the camera and light parameters and $M$ a human mask that can be estimated by the accumulated transmittance $M = \exp\left(-\sum_{j=1}^{N_i - 1} \sigma(\mathbf{x}_j)\delta_j\right)$. We simply repeat the adversarial patch on the texture stacks, which have shown to be able to improve the attack robustness (Hu et al., 2022). Because the texture stacks are disentangled, we can easily modify any part of the human body. Then we distort the texture stacks through Thin Plate Spline (TPS) (Hu et al., 2023) to simulate cloth distortions. Finally, we use the grid sampling function $g(\cdot, \cdot)$ to sample pixel colors from the modified texture stack according to the IUV map $x_{\text{IUV}}$ and combine it with background image $x_b$.

**Adversarial Patch Generation** Diffusion models have been found to be able to enhance the transferability of adversarial examples (Xue et al., 2023; Lin et al., 2023; Chen et al., 2023b). However, previous work either modifying the intermediate latent (Chen et al., 2023a;b) or diffusing and denoising an initial image (Lin et al., 2023; Xue et al., 2023), which has limited noise levels, resulting in impractical for physical adversarial attacks. UV-Attack contaminates the initial, highly noisy latent variable $z_T$ of a pre-trained stable diffusion model (Rombach et al., 2022) and remove classifier-free guidance. These approaches facilitate more substantial modifications and explore

broader latent spaces and less familiar areas, resulting in significantly higher physical attack success rates while maintaining transferability. However, due to the larger optimization space, relying solely on gradient-based algorithms may lead to local minima. To enhance optimization efficiency, we combined black-box and white-box algorithms. Initially, we employed the Particle Swarm Optimization (PSO) algorithm to find a suitable starting point. Subsequently, we utilized the Adam optimizer to fine-tune this point to further decrease the loss. To exclude the influence of stable diffusion prompts, we evaluated the average ASR by using different class labels except "person" in the COCO dataset as prompts.

**The Expection over Pose Transformation Loss**    Suppose the victim detector $D$ takes the rendered image $x$ as input and outputs a set of bounding boxes $b_i^{(x)}$ with confidence $\text{Conf}_i^{(x)}$ and class label $y$, we define the detection loss as $\mathcal{L}_{det} = \text{Conf}_{i^*}^{(x)}$, where $i^* = \arg\ \max_i \text{IoU}(\text{gt}^x, b_i^x)$. The adversarial loss $\mathcal{L}_{adv}$ can be expressed as the expectation over sampled pose parameters ($\boldsymbol{\theta}$), camera parameters ($\boldsymbol{\eta}$), light parameters ($\boldsymbol{\tau}$), and background images ($x_b$). The objective function is

$$\min_{z_T} \mathcal{L}_{adv} := \mathbb{E}_{\boldsymbol{\theta}\sim\Theta, \boldsymbol{\eta}\sim H, \boldsymbol{\tau}\sim T, x_b\sim B} \left[\mathcal{L}_{det}(x|z_T, \boldsymbol{\theta}, \boldsymbol{\eta}, \boldsymbol{\tau}, x_b)\right] \tag{5}$$

By minimizing $\mathcal{L}_{adv}$ with respect to $z_T$, we find a texture that makes the detector fail under a variety of realistic scenarios. Sampling over the camera and light parameters, including the camera and light positions and light intensity, can improve the attack success rate in real-world situations.

The final attack algorithm is shown in Algorithm 1.

---

**Algorithm 1** The `UV-Attack`

---

**Require:** Target detector $D$, pretrained latent diffusion model $\phi$, GMM parameters $\Theta$, background images $B$, number of PSO iterations $T_{\text{PSO}}$, number of PSO swarms $N_{\text{PSO}}$, number of Adam iterations $T_{\text{Adam}}$, pretrained UV-Volume model $M_{UV}$, batchsize $N_{batch}$.
 1: Initialize $N_{\text{PSO}}$ particle swarms with random positions $z_i$ and velocities $v_i$
 2: **for** $t = 1, \dots, T_{\text{PSO}}$ **do**
 3:     Update velocities $v_i^{(t+1)} = \omega v_i^{(t)} + c_1 r_1(p_i^{(t)} - z_i^{(t)}) + c_2 r_2(g^{(t)} - z_i^{(t)})$
 4:     Update positions $z_i^{(t+1)} = z_i^{(t)} + v_i^{(t+1)}$
 5:     Evaluate fitness ($-\mathcal{L}_{adv}$) of particles and update $p_i^{(t)}$ and $g^{(t)}$
 6: **end for**
 7: Initialize $z_T$ with the best position found by PSO: $z_T \leftarrow g^{(T_{\text{PSO}})}$
 8: **for** $t = 1, \dots, T_{\text{Adam}}$ **do**
 9:     Generate $x_{adv}$ through denoising $x_{adv} = Decoder(R_\phi(\dots R_\phi(R_\phi(z_T, T), T-1)\dots, 0))$.
10:     Sampling $\theta \sim \Theta, \boldsymbol{d} \sim D, \boldsymbol{\tau} \sim T, x_b \sim B$ with number $N_{batch}$
11:     Generate $N_{batch}$ image through Eq. 4.
12:     Compute adversarial loss through Eq. 5.
13:     Update $z_T$ using Adam optimizer
14: **end for**
15: **return** Adversarial patch $x_{adv}$

---

## 4 EXPERIMENTS

### 4.1 EXPERIMENTAL SETTINGS

**Training Details.**    We evaluate the success rate of digital attacks on the ZJU-Mocap dataset (Peng et al., 2021). For the physical attack, we collected videos of five individuals and used SPIN and DensePose to extract SMPL parameters and pseudo-supervised IUV maps for training the UV-Volumes. The model training and attack implementation are conducted on a single Nvidia 3090 GPU. We utilize a pretrained stable-diffusion model finetuned on 256×256 images. The diffusion process was set to 10 steps. The Particle Swarm Optimization (PSO) ran for 30 epochs with 50 swarms, and the Adam optimizer ran for 300 epochs. We collected 100 different backgrounds from both indoor and outdoor scenarios. For each epoch, we randomly sampled 100 poses, camera and light positions, and backgrounds for training. The camera is sampled from azim $\in [-180°, 180°]$ and elev $\in [0°, 30°]$.

Table 1: Comparison of different methods' ASR (%) on the multi-pose datasets using FastRCNN and YOLOv3. The confidence threshold $\tau_{conf}$ is set as 0.5.

| Victim Models | FastRCNN | | | | YOLOv3 | | | |
|---|---|---|---|---|---|---|---|---|
| $\tau_{IoU}$ | IoU0.01 | IoU0.1 | IoU0.3 | IoU0.5 | IoU0.01 | IoU0.1 | IoU0.3 | IoU0.5 |
| AdvYolo | 15.05 | 16.20 | 16.45 | 16.40 | 13.25 | 14.05 | 14.20 | 14.30 |
| AdvTshirt | 10.20 | 11.30 | 12.15 | 12.50 | 8.35 | 10.30 | 10.35 | 10.50 |
| NatPatch | 12.40 | 13.20 | 14.35 | 15.20 | 10.50 | 11.40 | 12.55 | 12.60 |
| AdvTexture | 1.50 | 6.75 | 10.05 | 17.90 | 1.40 | 8.40 | 14.50 | 18.10 |
| AdvCamou | 24.40 | 25.50 | 25.60 | 28.50 | 22.50 | 23.40 | 23.60 | 24.00 |
| Diff2Conf | 10.30 | 12.60 | 12.65 | 14.25 | 19.20 | 20.00 | 20.05 | 20.35 |
| DiffPGD | 12.40 | 12.70 | 12.80 | 13.40 | 15.10 | 17.50 | 17.80 | 19.40 |
| LDM | 24.50 | 25.40 | 25.45 | 25.50 | 26.40 | 27.50 | 28.30 | 28.45 |
| Ours$_{video}$ | 82.50 | 82.50 | 84.30 | 85.60 | 80.25 | 82.10 | 83.40 | 84.20 |
| Ours$_{GMM}$ | **85.65** | **85.80** | **86.65** | **92.75** | **84.50** | **88.40** | **89.50** | **90.40** |

**Victim Models and Evaluation Metrics.** To evaluate the attack success rate transferability of our attack, we trained the adversarial textures on FastRCNN (Girshick, 2015) and YOLOv3 (Redmon & Farhadi, 2018) respectively, and tested the transferability on a variety of the person detectors, including MaskRCNNHe et al. (2018), Deformable DETR (Zhu et al., 2020), RetinaNet (Lin et al., 2018), SSD (Liu et al., 2016), FCOS(Tian et al., 2019) and the most recent YOLO variant, YOLOv8 (Jocher et al., 2023). To evaluate the effectiveness of our attack, we measure the attack success rate (ASR) in both the digital world and the physical world. Most previous attacks set the IoU threshold $\tau_{IOU}$ as 0.5, which overestimated the attack effectiveness. In this work, we test the ASR under different IoU thresholds from 0.01 to 0.5. To exclude the influence of stable diffusion prompts, we evaluated the average ASR by using all class labels except "person" in the COCO dataset as prompts. This approach ensures that the ASR is not biased by specific prompts and provides a more generalized measure of the attack's effectiveness.

**Baseline attacks.** We compared our attack with different person detection attacks, including AdvYolo (Thys et al., 2019), AdvTshirt (Xu et al., 2020), AdvTexture (Hu et al., 2022), AdvCamou (Hu et al., 2023), and the latest diffusion-model-based attacks, including Diff2Conf (Lin et al., 2023) and DiffPGD (Xue et al., 2023). We also compared our method with a GAN-based attack, NatPatch (Hu et al., 2021). AdvTexture (Hu et al., 2022) and AdvCamou (Hu et al., 2023) are both texture-based attacks that can be directly compared. For AdvTshirt, Diff2Conf, and DiffPGD, which only add a single patch on the T-shirt, the attack performance drops significantly when the viewing angles change. Therefore, we tiled the patches produced by them onto the texture stacks for fair comparisons.

## 4.2 DIGITAL ATTACK RESULTS

**Evaluation on the Unseen Pose Dataset** (A) **Dataset and baseline settings.** To evaluate the ASR of different attacks under unseen pose scenarios, we randomly sample 1000 poses from the GMM and combine them with different backgrounds and camera and light conditions to construct an unseen pose dataset. Then we evaluate the average ASR on five different individuals from the ZJU-Mocap dataset, as shown in Figure 6. Because previous attacks did not consider the multi-pose scenarios, they cannot be directly evaluated. Therefore, we combine their methodologies with the UV-Volume model to generate images under different poses for testing. For AdvCamou (Hu et al., 2023), we first train a Voronoi diagram using their methods and then apply it to the texture stacks under a *fixed* pose. Then we test the generated Voronoi diagram on the multi-pose dataset. For other patch-based attacks, we use a fixed pose and put the generated patch on the texture stacks for training, followed by testing on the multi-pose dataset. Additionally, we added two additional baselines. One is to use the latent diffusion model with fixed poses to train the adversarial textures, referred to as the LDM attack. The other is to sample frames from a multi-pose video and then extract the pose parameters for the UV-Volume, referred to as Ours$_{video}$. (B) **Results analysis**. We present the test ASRs under different IoU thresholds in Table 1. All attacks were trained and tested on the same detectors . It indicates that our attack significantly outperforms previous attacks under unseen pose scenarios. Additionally, our attack maintains consistently high ASRs as the IoU thresholds decrease, in contrast to some previous attacks like AdvTexture, which significantly drops in performance. Experimental results demonstrate

Table 2: The white-box ASR (%) and transferability of different attacks. The IOU threshold is set as 0.1 and the confidence threshold is set as 0.5. Numbers with underline are white-box attacks. We **disable** pose sampling in the training and test process in this table for fair comparison.

| Method | Victim Models | | | | | | | |
|--------|--------|--------|-------|-------|--------|------|-----|--------|
| | FRCNN | YOLOv3 | DDETR | MRCNN | Retina | FCOS | SSD | YOLOv8 |
| AdvTshirt | 61.04 | 17.02 | 12.90 | 56.10 | 25.00 | 24.00 | 15.05 | 15.20 |
| AdvTexture | 22.54 | 10.01 | 9.05 | 15.30 | 8.00 | 15.60 | 8.24 | 6.50 |
| AdvCamou | 97.20 | 28.50 | 67.50 | 92.20 | 45.65 | 35.20 | 20.24 | 25.25 |
| Diff2Conf | 31.45 | 12.10 | 10.05 | 27.15 | 19.34 | 10.50 | 11.45 | 7.90 |
| DiffPGD | 35.37 | 16.24 | 13.40 | 29.50 | 16.18 | 12.43 | 14.29 | 10.50 |
| Ours | **98.36** | **50.45** | **68.18** | **93.50** | **75.95** | **64.50** | **38.50** | **49.50** |
| AdvTshirt | 11.03 | 75.00 | 15.44 | 8.20 | 18.32 | 15.40 | 10.05 | 16.50 |
| AdvTexture | 16.54 | 45.89 | 10.55 | 13.28 | 11.30 | 11.40 | 5.50 | 6.40 |
| AdvCamou | **24.40** | 93.18 | 16.38 | 22.57 | 25.40 | 22.66 | 25.40 | 18.30 |
| Diff2Conf | 15.45 | 42.10 | 10.20 | 18.40 | 15.20 | 6.40 | 12.47 | 10.02 |
| DiffPGD | 14.37 | 46.40 | 10.40 | 16.40 | 18.08 | 5.20 | 10.40 | 9.55 |
| Ours | 23.26 | **97.21** | **48.03** | **31.60** | **35.80** | **37.40** | **39.25** | **58.55** |

that our method can significantly improve attack resilience against pose variations. In addition, it is shown that sampling pose parameters from a GMM distribution rather than from video frames can improve the ASR. This is because we can synthesize unseen poses for training.

**Transferability** (A) We trained our attacks on FastRCNN and YOLOv3, respectively, and tested the transferability on various person detectors. Because AdvCamou is based on a static 3D model, we disable pose sampling in the training and testing in this table for fair comparisons. As shown in Table 2, our attack trained on FastRCNN has high transferability on Deformable-DETR, MaskRCNN, and RetinaNet. It is also shown that previous diffusion-based attacks, Diff2Conf and DiffPGD, have very low ASRs in our attack settings, even on the white-box models. We think this is because only adding noise to an initial image and then denoising it can not fully explore the LDM's latent space and, therefore, cannot find an optimum latent. In addition, our attack achieves high ASRs on the latest YOLOv8 detector (Jocher et al., 2023), while previous attacks did not perform very well on it.

## 4.3 Physical Attack Results

To evaluate the physical ASR, we first capture videos from targets and then estimate the pose and camera parameters to train the UV-Volume. In contrast to AdvCamou which requires calibrated 3D meshes, our approach only needs a short video of the target, thus simplifying the attack process. We find that the trained textures are transferable to different people, thereby allowing us to fine-tune the latent $z_T$ trained on the ZJU-Mocap datasets instead of training from scratch, which greatly reduces the training time. We set the finetuning epoch as 100 in the physical attacks. Then we print the textures on fabric materials and manufacture long sleeves and pants. As shown in Figure 1, our attack is robust to changes in pose, view, and distance. For quantitative analysis, we compute the ASR through $ASR = 1 - N_{succeed}/N_{total}$, where $N_{succeed}$ and $N_{total}$ represent the number of succeeded and total frames, respectively. We trained the texture on FastRCNN and tested the real-world ASRs on multiple detectors. The physical attack success rate with different detectors and confidence thresholds is shown in Figure 3. We achieved a high success rate on FastRCNN (0.82), MaskRCNN (0.78), and Deformable DETR (0.65) when $\tau_{conf}$ is 0.5 and is consistent when $\tau_{conf}$ changes, while RetinaNet, FCOS and YOLOv8 are relatively sensitive to $\tau_{conf}$'s change. The ASR for SSD is relatively low, which can be attributed to their fundamentally different model architectures.

Table 3: Attack performance on different body parts.

| Body Parts | UpperBody | UpperBody+Arms | UpperBody + Thighs | UpperBody+Arms+Thighs |
|------------|-----------|----------------|--------------------|-----------------------|
| RCNN | 23.26 | 47.38 | 56.39 | 84.35 |
| YOLOv3 | 41.16 | 64.67 | 66.93 | 87.68 |

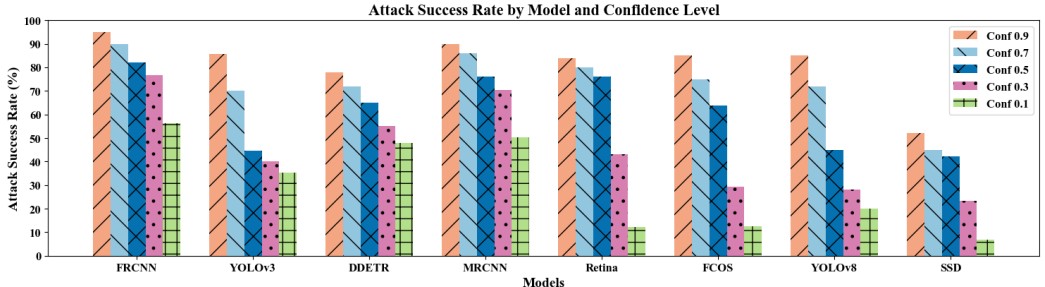

Figure 3: Physical attack success rates with different detectors and confidence thresholds. The attack is trained on the FastRCNN model, and the IoU threshold is set as 0.5.

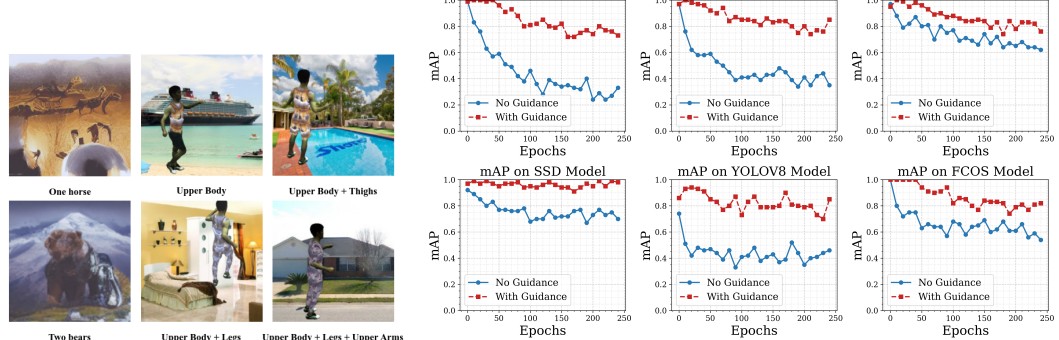

Figure 4: Visualization of adversarial textures and rendering results for different body parts.

Figure 5: The impact of classifier-free guidance on the mAP (lower is better). The adversarial patch is trained on the FastRCNN model. The IoU threshold is set as 0.5. It is shown that without the guidance (the blue line), the mAP drops more significantly.

## 4.4    ABLATION STUDY

**The influence of modifying different body parts.** Benefiting from the disentangled texture stacks, we can render arbitrary body parts, as shown in Figure 4. We evaluate the ASR of modifying different body parts and present these results in Table 3. As the coverage area of the adversarial textures increases, the attack success rate also progressively increases.

**The influence of classifier-free guidance**. We tested the impact of classifier-free guidance on the ASRs. We plot the mAP with and without classifier-free guidance on the white-box FastRCNN and the black-box objection detection models as the epochs progress, as shown in Figure 5. We observe that the diffusion model without classifier-free guidance can achieve lower mAP on the white-box and black-box models than using classifier-free guidance. We think this is due to the truncation effect, which normalizes the latent representations within the high-probability areas to improve image quality. This normalization restricts the generated images' noise levels and also reduces the transferability of untargeted attacks.

Table 4: Physical ASR in different scenes

|       | Outdoor |      |      | Indoor    |      |        |
|-------|---------|------|------|-----------|------|--------|
| Scene | 10am    | 2pm  | 6pm  | Classroom | Home | Office |
| ASR   | 0.94    | 0.98 | 0.92 | 0.94      | 0.92 | 0.96   |

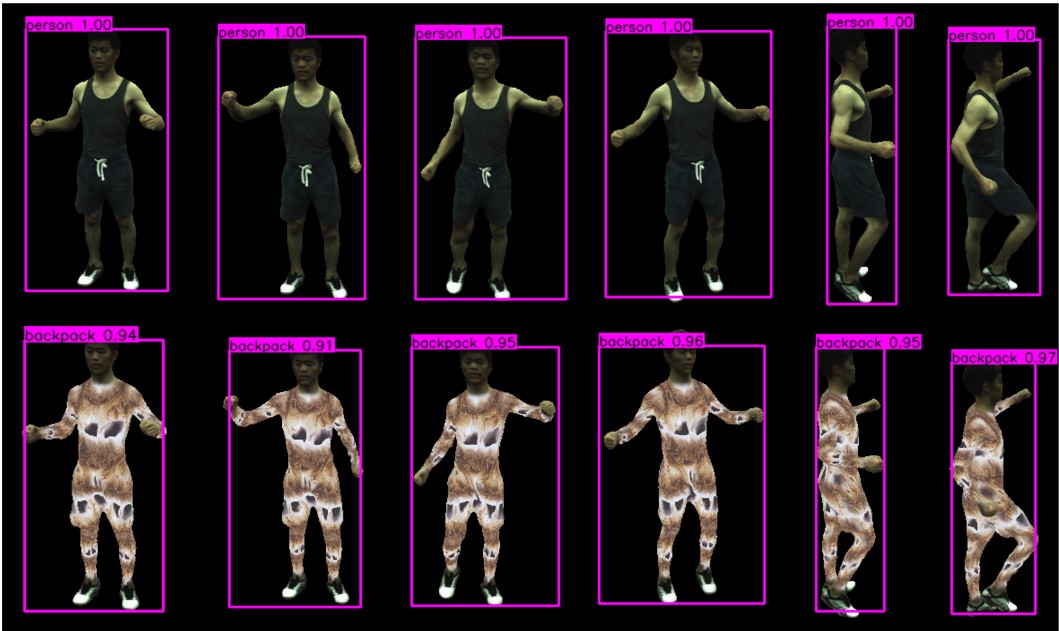

Figure 6: Compare with normal images and adversarial images in digital attacks.

**The influence of physical environment.** To test the Attack Success Rate (ASR) under more complex real-world environments, we compared the effectiveness of our attack in both indoor and outdoor scenes. For indoor scenes, data were collected from three different locations: a classroom, a home, and an office. For outdoor scenes, three videos were captured at 10 am, 2 pm, and 6 pm. Videos were recorded from various distances (1-5 meters) and included different poses, with 100 frames extracted from each video. The results are presented in Table 4, demonstrating that our attacks are effective across different scenes. We used RCNN as both the training and victim model.

**The influence of different NeRF models.** We compare the time needed to edit the clothing textures for different dynamic NeRF models (Weng et al., 2022; Geng et al., 2023; Shao et al., 2024). Our method, based on the UV-Volume, has the fastest speed among previous methods.

Table 5: Time required to edit clothing textures for different dynamic NeRF models

| Method | Control4D | HumanNeRF | Fast-HumanNeRF | Ours |
|---|---|---|---|---|
| Render time for one image | ∼5h | ∼2h | ∼15min | <0.5s |

## 5 CONCLUSION

We introduce `UV-Attack`, a novel physical adversarial attack leveraging dynamic NeRF-based UV mapping. Unlike previous methodologies that struggled with dynamic human poses, `UV-Attack` computes the expectation loss on the pose transformations, making our attack achieve high ASR to variable human poses and movements. The use of UV maps to directly edit textures without the need for gradient calculations of the NeRF model significantly enhances attack feasibility in real-world scenarios. Our experimental results show that `UV-Attack` outperforms the state-of-the-art method by large margins on unseen pose datasets, achieving high ASRs in diverse and challenging conditions. Ultimately, our attack not only advances the research of person detection attacks but also introduces a new methodology for real-world adversarial attacks on non-rigid 3D objects.

ACKNOWLEDGE

This work was supported in part by HK RGC GRF under Grant PolyU 15201323. Additionally, we would like to thank Wenxuan Zhang at Hong Kong Polytechnic University for his assistance with the ablation experiments.

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

## A ADDITIONAL EXPERIMENT RESULTS

### A.1 ATTACK RESULTS ON MORE OBJECT DETECTION MODELS

To evaluate the transferability of our attack, we trained the adversarial textures on FastRCNN (Girshick, 2015) and tested their transferability across a variety of person detectors with diverse backbones. These include models such as Deformable DETR (Zhu et al., 2020), RetinaNet (Lin et al., 2018), SSD (Liu et al., 2016), FCOS (Tian et al., 2019), as well as more recent detectors like RTMDET (Lyu et al., 2022), YOLOv8 (Jocher et al., 2023), ViTDET (Li et al., 2022), and Co-DETR (Zong et al., 2023). Notably, Co-DETR achieves the highest mean Average Precision (mAP) of 66% on the COCO object detection task. ViTDET, which employs the Vision Transformer as its backbone, allows us to further assess the attack's effectiveness across different backbone architectures. Co-DETR also supports different backbones. Our experiment tests Co-DETR on two different backbones, ResNet50 and Swin Transformer. We implement Co-DETR and ViTDET using the MMDET toolbox (Chen et al., 2019).

To evaluate the attack effectiveness, we adopt two different metrics, ASR and mAP. The ASR is defined as:

$$\text{ASR} = 1 - \frac{1}{|\mathcal{D}|} \sum_{i \in \mathcal{D}} \mathbb{I}\left(\exists b_i \in B_i : \text{IoU}(b_i, g_i) > \tau_{\text{IoU}} \land \text{Conf}(b_i) > \tau_{\text{Conf}} \land \text{label}(b_i) = \text{person}\right) \quad (6)$$

where $|\mathcal{D}|$ is the total number of images in the dataset, $B_i$ is the set of predicted bounding boxes for image $i$. The $\mathbb{I}(\cdot)$ is the indicator function that returns 1 if the condition inside is true, and 0 otherwise. In the experiment, we set both $\tau_{\text{IoU}}$ and $\tau_{\text{Conf}}$ as 0.5. Since ASR is influenced by the confidence threshold ($\tau_{conf}$), we use another metric, mAP, to measure the strength of the attack. mAP is calculated as the area under the Precision-Recall curve across different thresholds. The greater the drop in mAP, the more effective the attack is.

We compare our methods with two state-of-the-art person detection attacks (Adv-Camou (Hu et al., 2023) and DAP (Guesmi et al., 2024)) across 250 epochs. Figure 7 presents the ASR results across different models. We trained the adversarial patch using Fast-RCNN and evaluated its transferability to other detectors. The results demonstrate that our method significantly outperforms both Adv-Camou and UV-Attack in terms of transferability on a variety of detectors with different backbones. For instance, our approach achieves an ASR of 0.6 on the ViTDET model, which uses the Vision Transformer (ViT) as its backbone. Additionally, our attack shows a relatively low ASR on the state-of-the-art Co-DETR-SWIN model, indicating that the Co-DETR-SWIN model is more robust against adversarial examples, but still outperforms DAP and Adv-Camou. Figure 8 shows the mAP results across different models. It is shown that although DAP can quickly reduce mAP, its transferability is significantly lower than our method, especially on models with different architectures, such as Retina, SSD, FCOS, DETR and ViTDET.

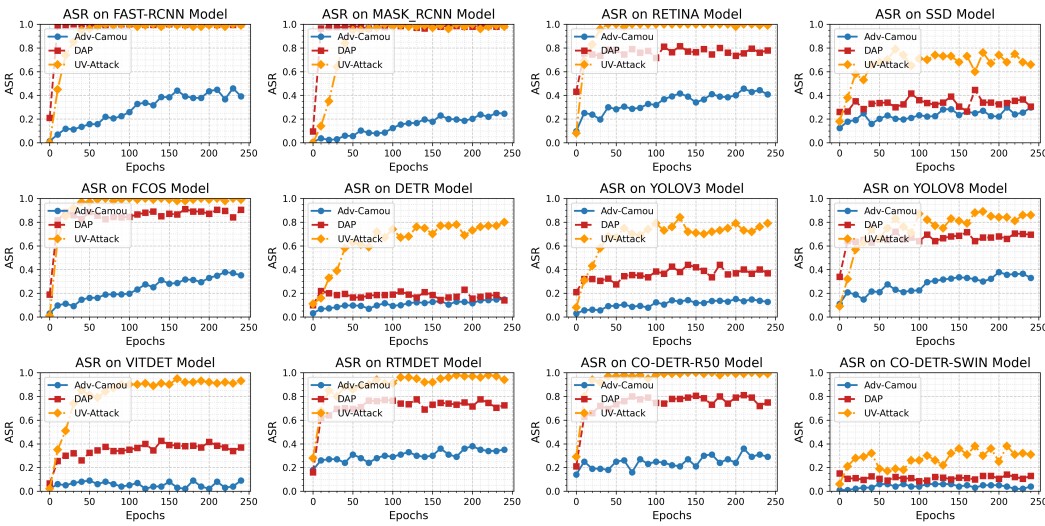

Figure 7: Comparison of the white-box ASR (on Fast-RCNN) and black-box ASRs (on the other nine detectors) of our proposed attack (`UV-Attack`) and two state-of-the-art person detection attacks (Adv-Camou (Hu et al., 2023) and DAP (Guesmi et al., 2024)) across 250 epochs. We attack a static person for a fair comparison. For DAP, we repeat the adversarial patch on clothes to make it cover the whole body. All adversarial patches are trained on the Fast-RCNN model.

# B ADDITIONAL ABLATION STUDIES

## B.1 THE INFLUENCE OF ENVIRONMENTS

To evaluate the attack performance in different environments, we test the attack success rate and mAP on a new indoor dataset (Quattoni & Torralba, 2009), which contains 67 Indoor categories, and a total of 15620 images. The number of images varies across categories, but there are at least 100 images per category. We select 13 common scenarios: bedroom, airport, office, etc. The attack results are shown in Figure 10. It is shown that all scenarios achieve a high ASR when the confidence threshold is set as 0.5. For the mAP, all scenarios are less than 0.2, and 6 scenarios are less than 0.1, including bedroom, bathroom, airport, office, library, and church.

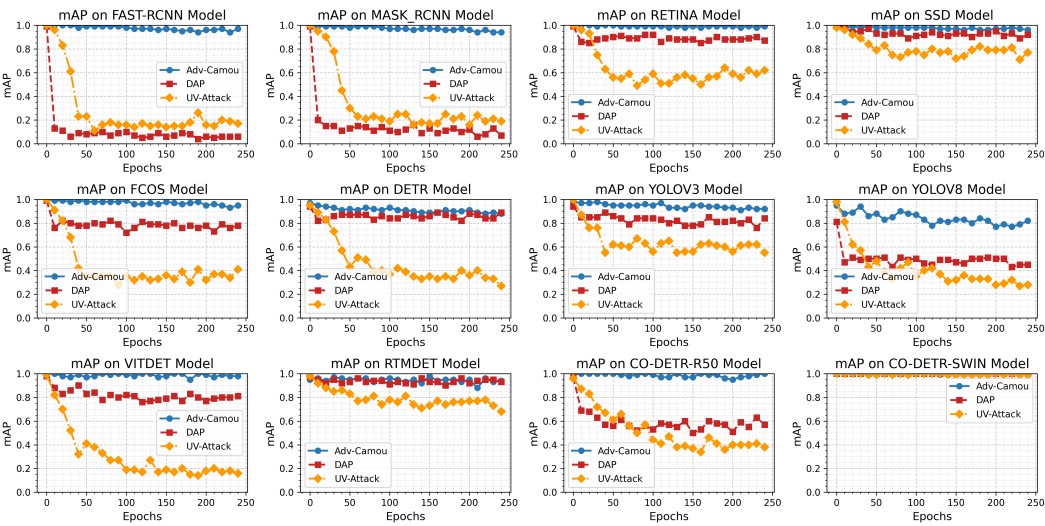

Figure 8: Comparison of the mAPs on different detectors of our proposed attack (`UV-Attack`) and two state-of-the-art person detection attacks (Adv-Camou (Hu et al., 2023) and DAP (Guesmi et al., 2024)) across 250 epochs. We attack a static person for a fair comparison. The adversarial patch is trained on the Fast-RCNN model. Compared with the previous attacks, our attack significantly decreases the mAP on different backbones, especially on ViTDET.

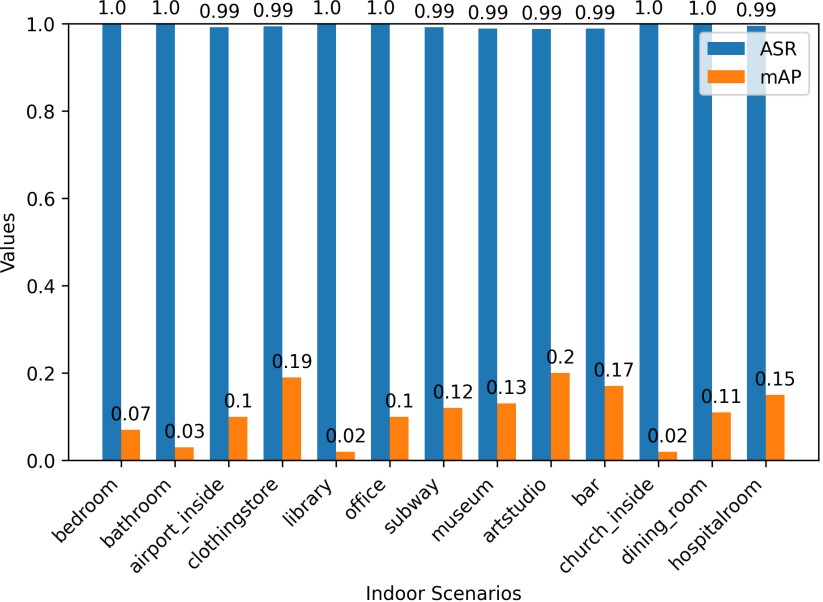

Figure 9: The attack success rate and mAP on a new indoor dataset (Quattoni & Torralba, 2009). It is shown that all scenarios achieve a high ASR when the confidence threshold is set as 0.5. For the mAP, we achieve mAPs of less than 0.1 on 6 out of 10 scenarios, including bedroom, bathroom, airport, office, library, and church.

## B.2 THE SENSITIVITY OF ASR TO DIFFERENT POSES

As shown in Figure 11, we analyze the sensitivity of ASR to different poses sampled from a GMM distribution. Specifically, 2000 examples were randomly sampled from a GMM with 10 components. The red line represents the number of examples corresponding to each component, while the blue bars indicate the ASR of the samples within each component. The results demonstrate that a high

ASR is consistently achieved across poses belonging to different components. For component 3, because no poses are sampled, the ASR is also zero.

Moreover, we evaluate the sensitivity of ASR to different poses sampled from a video in the ZJU-Mocap dataset, as shown in Figure 12. A total of 20 poses were selected by sampling every 20th frame from a video containing 400 frames. For each pose, 10 images were captured from different view angles. The results show that the ASR ranges from 0.76 to 0.92, with an average ASR of 0.82.

To explain why GMM can effectively model the distribution of poses, we plotted histograms of the 72-dimensional pose vector. Due to space constraints, we visualized dimensions 51–60. As shown in the Figure, most pose dimensions exhibit unimodal or bimodal distributions, which makes GMM an effective choice for modeling.

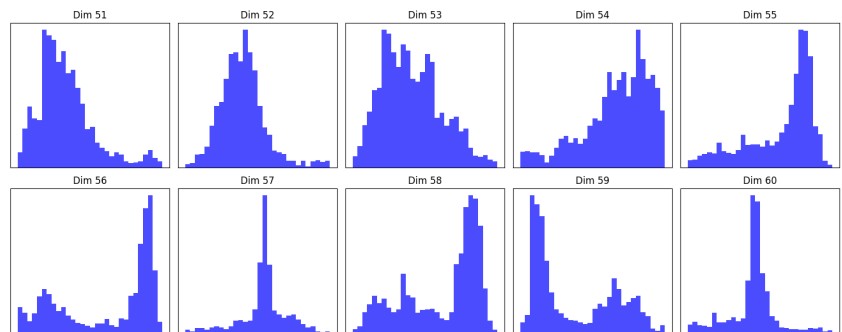

Figure 10: The distribution histograms of the pose vectors.

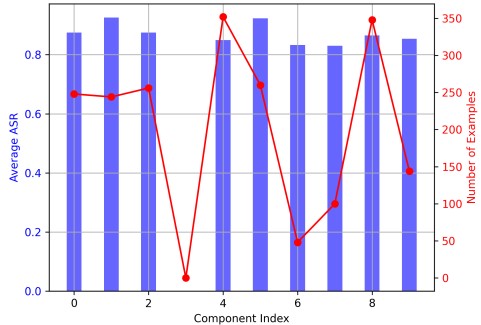

Figure 11: Illustration of ASR sensitivity to different poses sampled from a GMM distribution. A total of 2000 examples were randomly sampled from a GMM with 10 components. The red line shows the number of examples per component, while the blue bars represent the ASR for each component. High ASR values are observed across multiple pose components.

Figure 12: Illustration of ASR sensitivity to different poses sampled from a video from the ZJU-Mocap dataset. A total of 20 poses were sampled from a video with 400 frames with an interval of 20 frames. For each pose, we capture 10 images from different view angles. It is shown that the ASR varies from 0.76 to 0.92 with an average of 0.82.

### B.3 THE INFLUENCE OF THE GUIDANCE OF STABLE DIFFUSION MODEL

Figure 13 presents the generated patches with and without classifier-free guidance. The top row uses "Dog" as the prompt, while the bottom row uses "Horse" as the prompt. The results indicate that although classifier-free guidance produces more natural-looking images, it leads to a slower convergence of the adversarial loss. Interestingly, when using "Horse" as the prompt, the adversarial loss for the guidance setup is initially lower than for no guidance. We attribute this to the better

starting point provided by the guidance. However, as training progresses, the adversarial loss for the no-guidance setup decreases more rapidly, ultimately yielding a stronger attack.

Figure 20 presents comparisons of transferability under naturalness constraints. In this experiment, we incorporate classifier-free guidance to generate more natural adversarial patches. The results show that our method, even with naturalness constraints (UV-Attack-guidance), still outperforms previous state-of-the-art methods (e.g., DAP) in terms of transferability on models like Retina, SSD, and DETR. Additionally, our method demonstrates competitive performance on YOLOv3 and FCOS. This indicates that UV-Attack maintains strong transferability while achieving greater visual stealth through naturalness constraints.

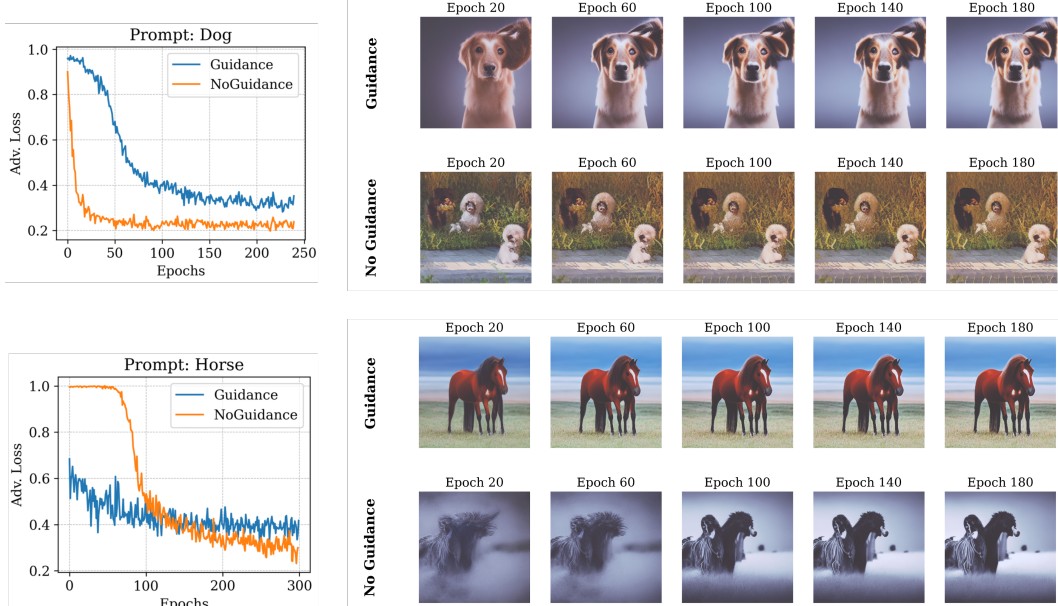

Figure 13: Visualization of generated patches with and without classifier-free guidance. The top row uses "Dog" as the prompt, and the bottom row uses "Horse" as the prompt. Although classifier-free guidance generates more natural-looking images, the adversarial loss converges more slowly. In the case of the "Horse" prompt, the adversarial loss with guidance starts lower than without guidance, likely due to a better initialization. However, as training progresses, the no-guidance setup achieves a faster decrease in loss, indicating a stronger attack.

### B.4 THE INFLUENCE OF IMAGE COMPLEXITY ON THE ATTACK EFFECTIVENESS

Figure 14 illustrates the changes in total variance (TV) and mAP as the number of training epochs increases. In the initial few epochs, TV increases rapidly and then stabilizes, while mAP continuously decreases, indicating that more complex images have a higher attack success rate. In Figure 12, we visualize the temporal changes of the adversarial patch. It can be observed that during the early stages of training, the image changes significantly, followed by little to no change. Particularly in the last row, the image starts off relatively simple, so the loss does not decrease. As the complexity increases, the loss quickly converges to a smaller value.

### B.5 THE ATTACK RESULTS ON ROBUST MODELS.

Figure 15 illustrates the attack results on the robust model. We test our attack on the RobustDet (Ziyi Dong, 2022). We use the VGG version of the pretrained RobustDet model. The RobustDet is adversarially trained on PGD and DAG attacks. We train the adversarial patch on Fast-RCNN and test the mAP on the black-box RobustDet. The results show that although RobustDet is resistant to PGD and DAG attacks, its mAP drops from 1.0 to 0.85 in our adversarial examples.

## B.6 THE INFLUENCE ON DIFFERENT PROMPTS

Figure 17 shows the training loss curves across epochs under different prompts. Although the convergence speed varies slightly among different prompts, they all converge to a relatively small value. This indicates that our method is effective across different prompts.

## B.7 THE TRANSFERABILITY OF ENSEMBLE METHOD

The ensemble method is widely used to improve transferability. However, we found that direct training on RCNN and YOLOv3 resulted in slow loss convergence. To address this issue, we first trained on RCNN for 300 epochs and then trained on both RCNN and YOLOv3 for an additional 20 epochs. Figure 18 illustrates the mAP degradation across different methods. The ensemble method achieved lower mAP on five out of six models, indicating that the ensemble approach effectively reduces mAP.

## B.8 THE TRANSFERABILITY ON UNSEEN SUBJECTS

Figure 19 shows the attack results on unseen subjects. We train the adversarial patch on subject 1 and then test the ASR on subject 2. The attack success rate decreased by 20%-35%. We believe this is due to significant differences in body shapes and motion habits between different subjects, which caused the drop in success rate.

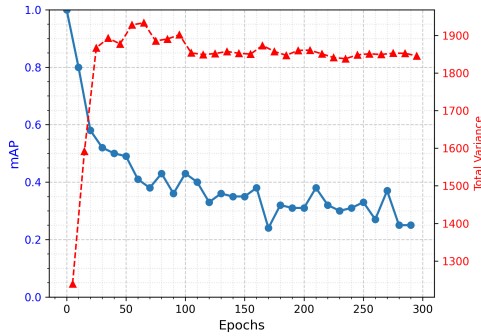

Figure 14: Illustrations on the changes in total variance (TV) and mAP as the number of training epochs increases. In the initial few epochs, TV increases rapidly and then stabilizes, while mAP continuously decreases, indicating that more complex images have a higher attack success rate.

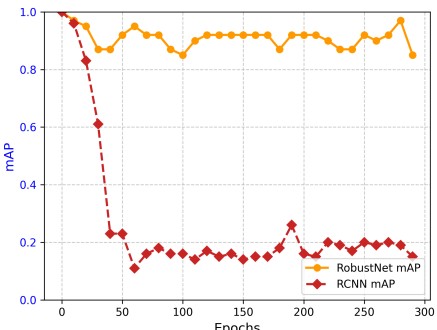

Figure 15: The attack results on the RobustDet (Ziyi Dong, 2022). We use the VGG version of the pretrained RobustDet model. The model is adversarially trained on PGD and DAG attacks. The results show that RobustNet is relatively resistant to our attack.

## C ADDITIONAL EXPERIMENTS IN THE PHYSICAL ENVIRONMENTS

We demonstrate the attack effects in three different scenarios: a hallway, a lawn, and a library. For each scenario, we recorded a video of approximately 1 minute with a frame rate of 30 FPS, using an iPhone 13 as the recording device. The confidence threshold is set as 0.5. The results are shown in Figure 16. The adversarial patch is trained on the Faster-RCNN model. In these three scenarios, we achieve ASR of 0.84, 0.83, and 0.80, respectively. The experiments show that we can successfully fool Faster-RCNN in various scenarios.

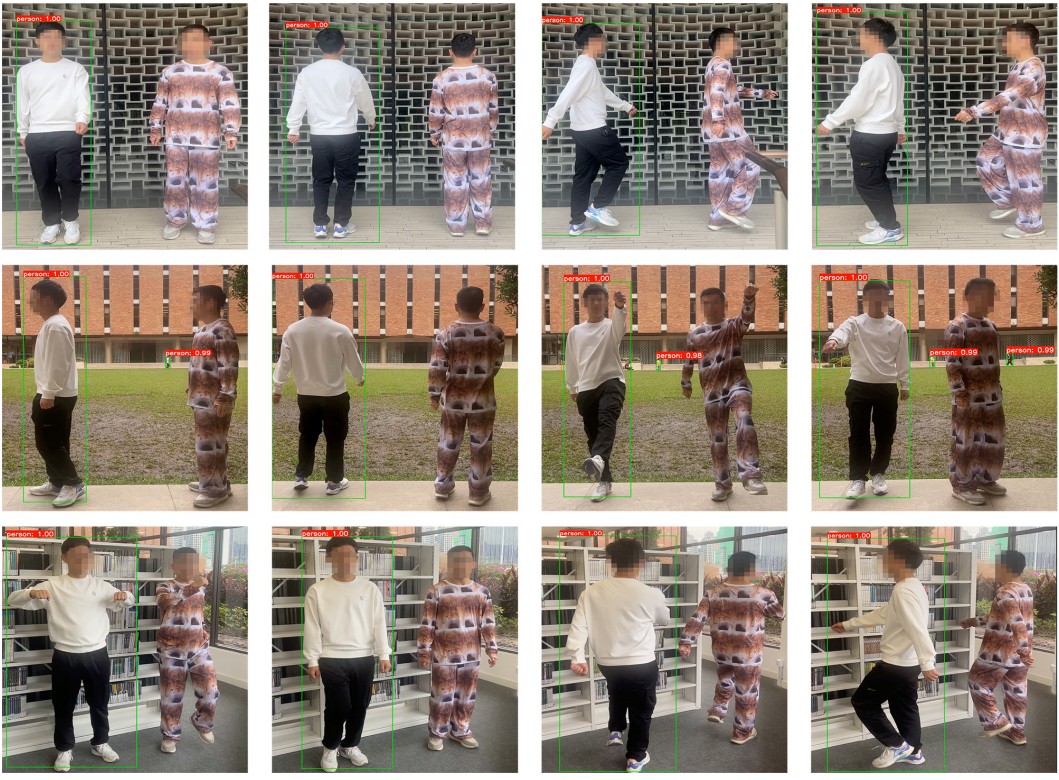

Figure 16: The attack effects are demonstrated in three different scenarios: a hallway, a lawn, and a library. For each scenario, we recorded a video of approximately 1 minute with a frame rate of 30 FPS, using an iPhone 13 as the recording device. The confidence threshold is set as 0.5. The experiment shows that we can successfully fool Faster R-CNN in various scenarios.

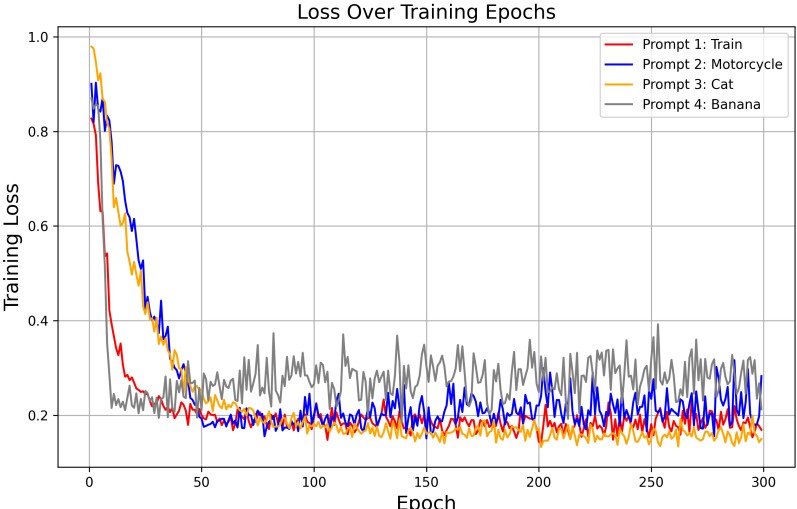

Figure 17: Visualization of the training loss changing with the epochs for different prompts.

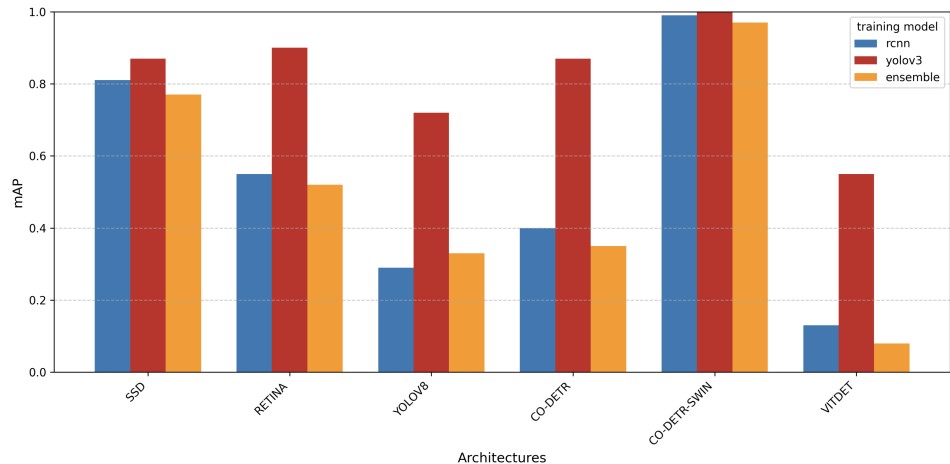

Figure 18: Comparison of the transferability on different white-box models. The ensemble method achieves lower mAP on 5 out of 6 black-box models. The Y-axis is the mAP.

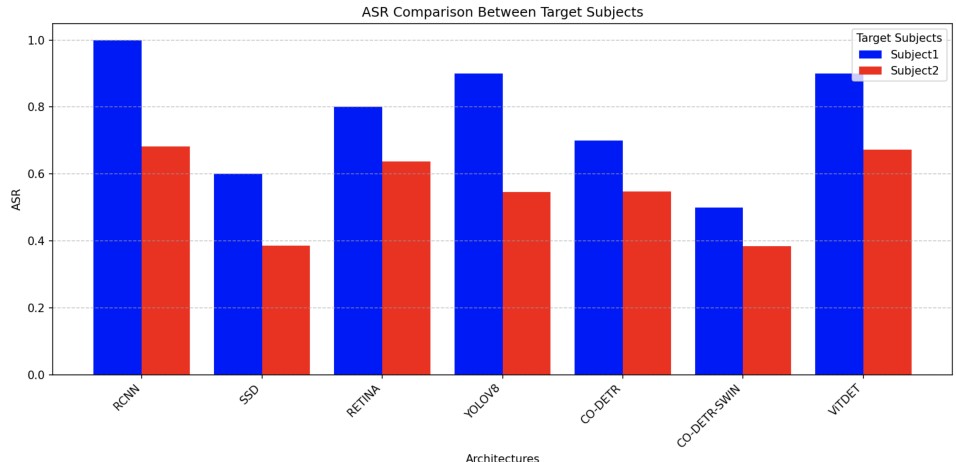

Figure 19: The attack transferability on unseen subjects. We train the adversarial patch on subject 1 and then test the ASR on subject 2. The white-box model is Faster-RCNN. The Y-axis is the attack success rate.

# D  VISUALIZATION OF ATTACK TRANSFERABILITY ON UNSEEN REAL-WORLD POSE SEQUENCES

We evaluate the transferability of our attack using video sequences from the ZJU-Mocap dataset, which were not seen during the training phase. Figure 21 compares the ASR of three methods—Random, DAP (Guesmi et al., 2024) and UV-Attack across five pose sequences (#386, #387, #390, #392, #394) from the ZJU-Mocap dataset. UV-Attack consistently achieves the highest ASR, peaking at 0.91 on #387, while DAP shows moderate performance and Random performs poorly. This highlights the effectiveness of UV-Attack on unseen datasets. We also visualize some results in Figure 22. The first column displays the original pose images. By extracting the pose parameters, we generate human images wearing normal clothing and adversarial clothing. The results demonstrate that our attack achieves a high success rate, with an average ASR of 79% across five unseen pose sequences, even on previously unobserved poses, significantly outperforming the SOTA DAP attack (Guesmi et al., 2024), which achieves an ASR of 38%. This highlights our approach's strong effectiveness and robustness in real-world scenarios.

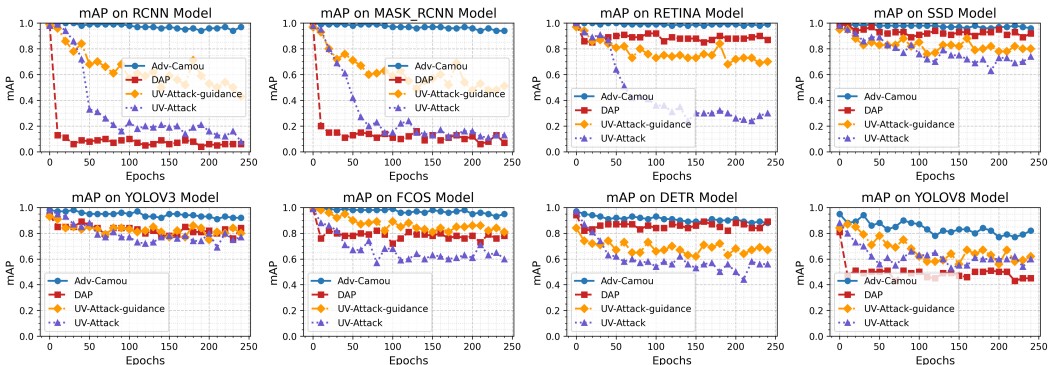

Figure 20: Comparisons of transferability with the naturalness constraints. Here we add the classifier-free guidance to generate natural images. It is shown that our method outperforms previous SOTA methods regarding transferability on Retina, SSD, and DETR and shows competitive results on YOLOv3 and FCOS.

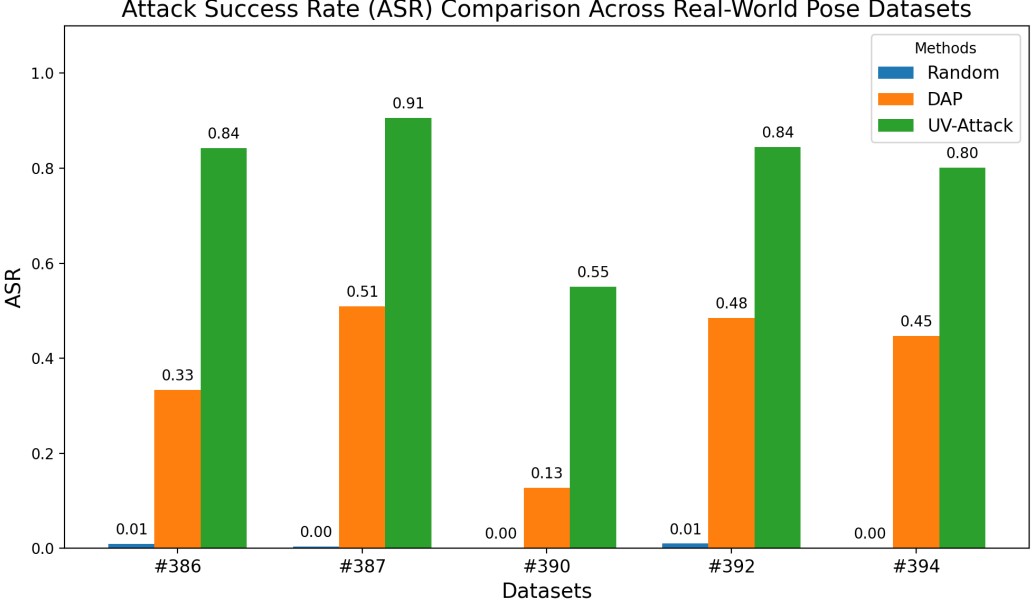

Figure 21: Comparison of ASR across different real-world pose sequences (#386, #387, #390, #392, #394) from the ZJU-Mocap for three methods: Random, DAP, and UV-Attack. All these pose sequences are unseen during the training. UV-Attack achieves the highest ASR consistently across all datasets, demonstrating its effectiveness compared to the other methods.

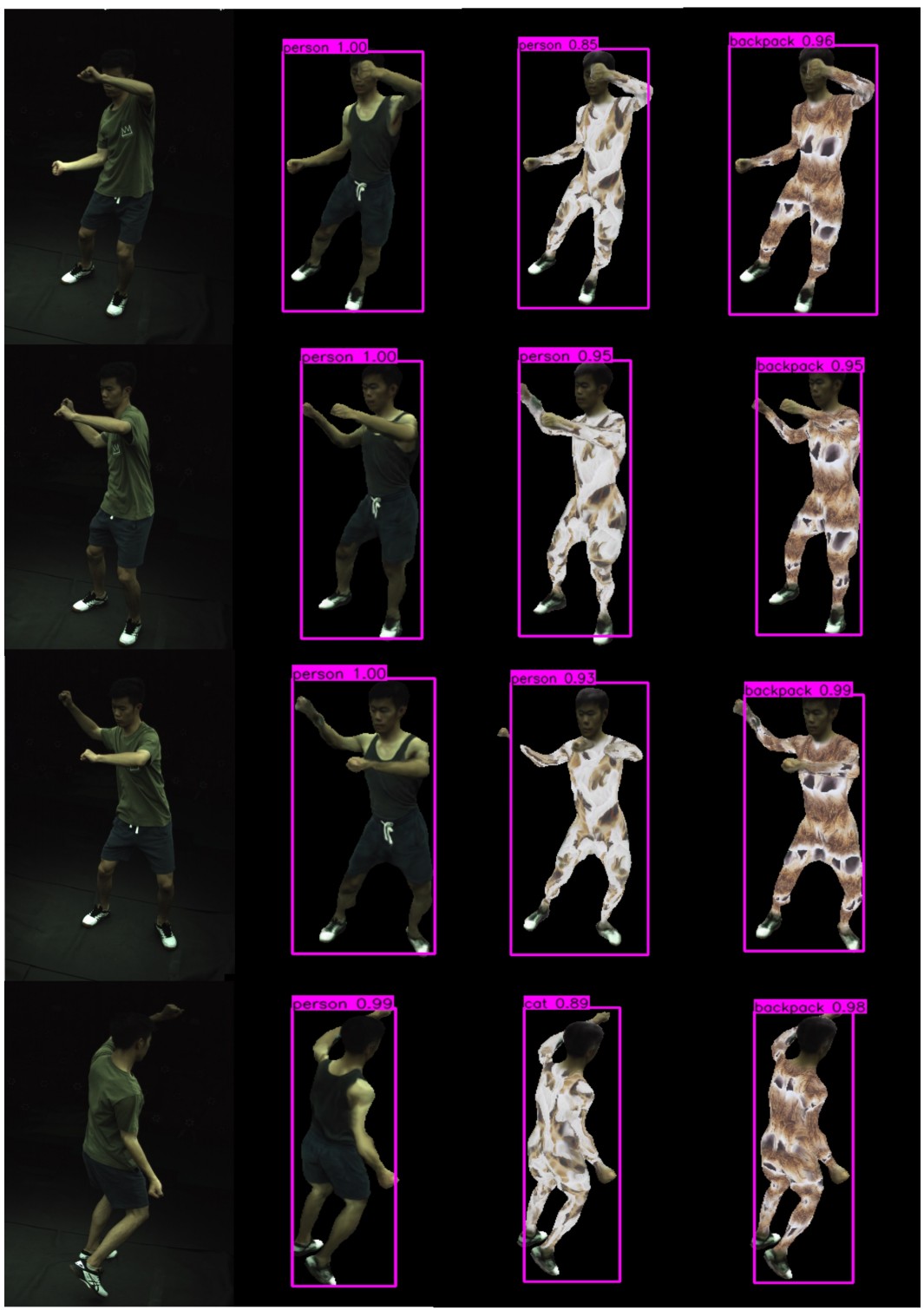

Figure 22: Visualization of attack results on unseen poses of DAP and our attack using video sequences from the ZJU-Mocap dataset, which were not seen during the training phase. The first column displays the original pose images. By extracting the pose parameters, we generate human images wearing normal clothing (shown in the second column). The third and fourth rows are the adversarial clothes generated by the DAP and UV-Attack. The results demonstrate that our attack achieves a higher success rate than previous SOTA DAP attacks, even on unseen poses, highlighting its effectiveness in real-world scenarios.

