# OpenReview forum: "UV-Attack: Physical-World Adversarial Attacks on Person Detection via Dynamic-NeRF-based UV Mapping"
_ICLR.cc/2025/Conference — ICLR 2025 Poster_

### Official Review · Reviewer_qDwz · 2024-11-01

**Soundness:** 3
**Presentation:** 3
**Contribution:** 2
**Rating:** 8
**Confidence:** 5

**Summary:**

The paper presents UV-Attack, a method designed to generate effective physical adversarial attacks targeting person detection systems by leveraging dynamic Neural Radiance Fields (NeRF) and UV mapping. This approach allows adversarial textures on clothing to adapt seamlessly to various human poses, addressing the challenges posed by non-rigid human movement. UV-Attack introduces two key components: a custom Expectation over Pose Transformation (EoPT) loss function to enhance attack success across diverse poses and viewpoints, and adjustments to the diffusion model to improve the transferability of adversarial examples to different detection systems.

**Strengths:**

- The paper introduces a novel application of dynamic NeRF and UV mapping, addressing the challenge of generating adaptable adversarial textures for non-rigid objects like human bodies.
- The authors propose an Expectation over Pose Transformation (EoPT) loss, which improves the patch’s robustness by ensuring its effectiveness across a wide range of poses.
- Extensive experiments across multiple person detection models and in various settings, including both white-box and black-box scenarios.
- The physical testing on printed clothing showcases UV-Attack’s applicability in real-world scenarios.
- The writing and presentation are clear and easy to follow.

**Weaknesses:**

- The paper focuses heavily on maximizing attack success rates against person detection models but does not address visual inconspicuousness or stealth. For a physical adversarial attack to be practical in real-world scenarios, it should evade not only machine detection but also appear natural and undetectable to human observers. UV-Attack’s textures might draw attention due to potentially unnatural patterns, especially in public settings where they could be easily noticed.

=> Action: I recommend introducing constraints that limit the patch’s appearance to more natural or common textures, to balance adversarial effectiveness with visual stealth. This could improve the applicability of UV-Attack in sensitive contexts, such as surveillance evasion, where inconspicuousness is essential.

- The authors report success rates for state-of-the-art attacks that differ significantly from those in the original publications. This inconsistency complicates direct comparisons and may impact the perceived credibility of UV-Attack’s comparative performance. Additionally, some state-of-the-art techniques apply constraints on patch appearance to ensure inconspicuousness, making comparisons with unconstrained approaches potentially unfair.

=> Action: To improve clarity and credibility, the authors should explicitly document the reasons behind any benchmarking deviations and, where possible, apply similar constraints to UV-Attack’s patches for fair comparison. Aligning evaluation protocols with prior works or noting significant differences in approach would enhance the validity of the comparison.

Less Important:

- The paper overlooks relevant related work, particularly “DAP: A Dynamic Adversarial Patch for Evading Person Detectors,” which similarly aims to create patches that are robust to non-rigid transformations. Including this work in the literature review would help contextualize UV-Attack’s contributions within the existing research.

=> Action: Discussing DAP and other similar methods would provide readers with a clearer understanding of UV-Attack’s novelty and advancements, especially in handling non-rigid transformations.

- The use of dynamic NeRF in UV-Attack is computationally intensive, which could limit applicability in scenarios requiring frequent re-designs of the patch for different targets or environments. While this is less relevant for single, offline generation, the computational demand could become a significant limitation if applications required ongoing adaptations.

**Questions:**

Check weakness

---

> ### Author Response · Authors · 2024-11-22
> **### Comments to Reviewer qDwz**
>
> We thank you for your reviews and address your concerns as follows.
>
> **Response to W1**: Thank you for this suggestion. We agree that introducing visual stealth constraints is crucial for improving the practicality of the method. In Appendix B.3 and Figure 13, we compared adversarial patches generated with and without classifier-free guidance. The results demonstrate that applying classifier-free guidance constraints produces images that appear more natural. However, as shown in Figure 5 and Figure 13, this approach comes with a trade-off: the training loss increases and the transferability decreases compared to patches generated without such constraints. For future work, we plan to modify the stable diffusion conditions to generate more natural and realistic clothing patterns while maintaining adversarial effectiveness, such as those inspired by Bohemian Fashion Style.
>
> **Response to W2**: As mentioned in L359-362, we train these adversarial patches on a single pose and then evaluate the ASR on a multi-pose dataset. Since previous attacks did not account for pose variations during their optimization process, they exhibit low ASR on unseen poses. This highlights the robustness of our method in handling diverse pose transformations.
>
> We compared the transferability of our method with previous approaches when constraints are introduced in Appendix B.3. Figure 20 presents comparisons of transferability under naturalness constraints. In this experiment, we incorporate classifier-free guidance to generate more natural adversarial patches. We train the adversarial patches on Faster-RCNN and then test on the ASR on 7 black-box models. The results show that our method, even with naturalness constraints (UV-Attack-guidance), still outperforms previous state-of-the-art methods (e.g., DAP) in terms of transferability on models like Retina, SSD, and DETR. Additionally, our method demonstrates competitive performance on YOLOv3 and FCOS.
>
> **Response to W3**. In Appendix A1, we compared our method with DAP in terms of ASR and mAP. The results show that while DAP can quickly reduce the mAP on a white-box model, its transferability is significantly lower than our method. This may be because DAP modifies adversarial patches at the pixel level, which tends to overfit to the white-box model.
>
> In Appendix D, we compare our method with DAP in terms of ASR on unseen real-world pose sequences. As shown in Figure 21 and Figure 22, our method achieves an average ASR of 79% on five unseen pose sequences, significantly outperforming DAP, which only has an average ASR of 38%.
>
> Furthermore, compared to DAP, which focuses on handling non-rigid clothing transformations, our method addresses not only clothing transformations but also pose transformations, which are far more complex. This broader scope makes our approach more robust in real-world scenarios.
>
> In addition, DAP's techniques can be integrated into our method. For instance, the crease transformation introduced in DAP (Eq. 4 in the DAP paper) could replace the TPS transformation (Eq. 4 in our paper), potentially enhancing our method further.
>
> **Response to W4**: With the rapid development of dynamic NeRF techniques, we expect the computational time and cost of finetuning a dynamic NeRF will continue decreasing, such as combining UV-Volume with the SHERF[1], making this approach more feasible in the future.
>
> Reference:
> [1]SHERF: Generalizable Human NeRF from a Single Image, ICCV2023

---

> > ### Author Response · Authors · 2024-11-25
> >
> > Dear Reviewer qDwz，
> >
> > Thank you very much for your valuable comments on our submission! We have clarified and responded to your questions, and would like to know if our responses satisfy you. We are happy to reply to any unclear parts, if any, since the last day for reviewers to ask questions to authors is approaching!
> >
> > Best, Authors

---

### Official Review · Reviewer_3gqc · 2024-11-02

**Soundness:** 3
**Presentation:** 3
**Contribution:** 2
**Rating:** 6
**Confidence:** 4

**Summary:**

This paper introduces UV-Attack, a novel adversarial approach using dynamic-NeRF-based UV mapping to achieve high ASRs on person detectors across diverse human actions and viewpoints. By generating UV maps rather than static RGB images, UV-Attack enables real-time texture modifications, making it practical and adaptable to unseen poses. Experiments show a 92.75% ASR against FastRCNN and 49.5% on YOLOv8, significantly surpassing prior methods and showcasing the potential of dynamic NeRF for effective adversarial attacks on moving human targets.

**Strengths:**

1. The idea of expanding the perturbation space by removing classifier-free guidance is interesting and brings a fresh perspective for boosting the transferability in the physical world.
2. The experiments are comprehensive, covering both digital and physical environments for thorough validation.
3. The paper is well-organized, with a clear and logical flow.

**Weaknesses:**

1. I’m a bit unclear on the Expectation over Pose Transformation (EoPT) loss. How does it differ from the standard Expectation over Transformation (EoT)? From Equation (5), it seems that the transformations traditionally applied at the image level have been shifted to include pose, camera, and lighting changes.
2.  Why do you choose YOLOv3 and Faster R-CNN as the target models rather than other, potentially more recent models?  Is there an underlying reason for this choice?
3. In line 431, the paper mentions that structural differences between models like SSD and the target model lead to limited transferability. Why not try a model ensemble approach to boost transferability, which is often used in physical attacks?
4. Could you clarify how the target model is set up? Is it just a pretrained model, or is it fine-tuned on a specific dataset? Additionally, how do each of the models perform on clean samples? This would serve as an important comparison, especially for physical-world testing. Since factors like distance, angle, and setting can greatly impact a detector’s performance in the physical world, poor performance on clean samples would reduce the significance of the attack itself.
5. For the physical experiments: (1) there is no mention of how many frames were captured in the video used to calculate ASR; if the frame count is too low, the results may lack credibility; (2) the physical-world attack examples are limited in the current draft—please provide a more complete set.

**Questions:**

Please refer to the weakness. If my questions are addressed, I will raise the score.

---

> ### Author Response · Authors · 2024-11-22
> **### Comments to Reviewer 3gqc**
>
> We thank you for your reviews and address your concerns as follows.
>
> 1. How does EoPT differ from the standard Expectation over Transformation (EoT)?
>
> Answer: Traditional EoT only includes 2D transformations, such as translation and rotation. In our proposed EoPT (Equation 5), we consider pose variations for the first time. Since pose variations cannot be simulated at the image level—especially when it comes to generating new, unseen poses—we sample in the SMPL parameter space in Equation 5, allowing us to generate novel poses.
>
> 2. Why do you choose YOLOv3 and Faster R-CNN as the target models rather than other, potentially more recent models? Is there an underlying reason for this choice?
>
> Answer: We use YOLOv3 and Faster R-CNN as the white-box models and evaluate their transferability against more recent models, including YOLOv8, as shown in Table 2. **Moreover, we test on additional modern models, including ViTDET, RTMDET, Co-DETR-R50, and Co-DETR-SWIN, in Appendix A1, as well as on one robust model, RobustDet, in Appendix B.5.** Compared with previous attacks, our attack significantly decreases the mAP on different backbones, including ViTDET, which uses a ViT transformer as its backbone.
>
> 3. In line 431, the paper mentions that structural differences between models like SSD and the target model lead to limited transferability. Why not try a model ensemble approach to boost transferability, which is often used in physical attacks?
>
> Answer: The ensemble method is widely used to improve transferability. However, we found that direct training on RCNN and YOLOv3 resulted in slow loss convergence. To address this issue, **we first trained on RCNN for 300 epochs and then trained on both RCNN and YOLOv3 for an additional 20 epochs.** Figure 18 in the Appendix illustrates the mAP degradation across different methods. The ensemble method achieved lower mAP on five out of six models, indicating that the ensemble approach effectively reduces mAP.
>
> 4. Could you clarify how the target model is set up? Is it just a pretrained model, or is it fine-tuned on a specific dataset? Additionally, how do each of the models perform on clean samples? This would serve as an important comparison, especially for physical-world testing. Since factors like distance, angle, and setting can greatly impact a detector’s performance in the physical world, poor performance on clean samples would reduce the significance of the attack itself.
>
> Answer: All models used in our experiments are pre-trained models, many of which were trained on the large-scale COCO dataset. These models are widely recognized for their strong performance in person detection, achieving high mAPs on real-world datasets. Among them, Co-DETR represents a state-of-the-art model for person detection. **We further assess the performance of these models on clean samples.** As shown in Figure 8, at the beginning of the experiments, the mAP (Mean Average Precision) of all models is close to 1. This demonstrates that the models can effectively detect humans in clean images, even under various backgrounds. Therefore, any subsequent drop in performance due to the attack can be attributed to the adversarial perturbations rather than inherent weaknesses in the models.
>
> 5. For the physical experiments: (1) there is no mention of how many frames were captured in the video used to calculate ASR; if the frame count is too low, the results may lack credibility; (2) the physical-world attack examples are limited in the current draft—please provide a more complete set.
>
> Answer: (1) We collected multiple videos in different scenarios, with each video lasting about one minute and recorded at 30 fps, resulting in approximately 1800 images per video. We then sampled one frame every 10 frames and calculated the attack success rate.
> (2) **We show more physical attack examples in Appendix C.** We demonstrate the attack examples in three different scenarios: a hallway, a lawn, and a library. For each scenario, we recorded a video of approximately 1 minute with a frame rate of 30 fps, using an iPhone 13 as the recording device. The experiments show that we can successfully fool Faster R-CNN in various scenarios.

---

> ### Author Response · Authors · 2024-11-25
>
> Dear Reviewer 3qgc，
>
> Thank you very much for your valuable comments on our submission! We have clarified and responded to your questions, and would like to know if our responses satisfy you. We are happy to reply to any unclear parts, if any, since the last day for reviewers to ask questions to authors is approaching!
>
> Best, Authors

---

> > ### Comment · Reviewer_3gqc · 2024-11-25
> >
> > Thank you for your response. My concerns have been addressed, so I decide to raise my rating.

---

> > > ### Author Response · Authors · 2024-11-27
> > >
> > > We are very grateful to the reviewers for recognizing our work. Your support is important for our efforts.

---

### Official Review · Reviewer_mhf9 · 2024-11-03

**Soundness:** 4
**Presentation:** 4
**Contribution:** 3
**Rating:** 8
**Confidence:** 5

**Summary:**

The authors propose a novel method to attack person detectors for various human poses in the physical world. They incorporate dynamic NeRF-based UV mapping and the Gaussian Mixture Model to sample and generate unseen human poses with editable textures. The texture is generated by a latent diffusion model. They then combine PSO and Adam optimizer to find the optimal diffusion latent variable for the adversarial texture. The adversarial texture outperforms multiple baselines in both the digital and the physical world.

**Strengths:**

1.	The writing is clear and easy to understand.
2.	The goal of this paper is important to this area, and the solution is technically sound.
3.	The experiments are comprehensive. The authors evaluate the adversarial effectiveness of the adversarial patterns under various parameters, including poses, viewing angles, and IoU thresholds. White box and transfer study are both included.

**Weaknesses:**

1.	The digital test setting seems a little problematic. The test poses are generated by GMM, which could be unreal. This raises the training-test contamination issue since a GMM-based model is also included in the model comparison. I suggested using videos from a different source.
2.	It lacks a null model for comparison: a non-adversarial pattern, such as an everyday clothes pattern, or a generated pattern from a random initial point.
3.	The training detail section is confusing; please see the questions.

**Questions:**

1.	What is the training dataset? Does it include the videos recorded by the authors? If so, the authors should make a split on the training and test dataset that consists of different subjects and backgrounds.
2.	Are the training datasets for digital and physical evaluation different? If so, why? What is the physical adversarial effectiveness of the digitally optimized patterns?
3.	Is the SMPL modeling for each subject the same? What's the transferability to unseen subjects? Does this mean that this method requires training a UV-volume model and optimizing for each person who is going to wear the adversarial clothes? If so, this method seems to have a great limitation; how to address this?

I'm happy to raise the score if these concerns are addressed.

---

> ### Author Response · Authors · 2024-11-22
> **### Comments to Reviewer mhf9**
>
> We thank you for your reviews and address your concerns as follows.
>
> 1. What is the training dataset? Does it include the videos recorded by the authors? If so, the authors should make a split on the training and test dataset that consists of different subjects and backgrounds.
>
> Answer: **For both digital and physical attacks, we split the training and test datasets to make them have different poses and backgrounds.** In the digital attacks, the dynamic 3D human model is from the ZJU-Mocap datasets. In the physical attacks, we recorded the target person's videos from different viewpoints and used them to train the UV-Volume model. The background dataset used in our paper is collected on the Internet, which includes over 100 indoor and outdoor images. We split it as a training and a test dataset. Moreover, in Appendix B1, we test our attack on a new background dataset, Indoor09. We select 13 common scenarios and achieve mAPs of less than 0.2 in all scenarios.
>
> 2. Are the training datasets for digital and physical evaluation different? If so, why? What is the physical adversarial effectiveness of the digitally optimized patterns?
>
> Answer: In the digital evaluation, we test the attack on ZJU-Mocap dataset. This is because due to the sensitivity of 3D data, collecting large-scale 3D human data independently may raise privacy concerns. Therefore, we opted to use the open-source dataset ZJU-Mocap, which contains over 10 high-quality human subjects and meets the requirements of our research.
>
> In the physical evaluation, we collected the target person's image and used it to train a UV-Volume model. This is because different person may have different body shapes. Therefore, we first generate the adversarial patch on the ZJU-Mocap dataset and then finetune the adversarial patch on the target person to improve the attack success rate. In Appendix C, we show the physical adversarial effectiveness of the digitally optimized patterns in three different scenarios. It is shown that our attack can consistently attack the detectors in different environments.
>
> 3. Is the SMPL modeling for each subject the same? What's the transferability to unseen subjects? Does this mean that this method requires training a UV-volume model and optimizing for each person who is going to wear the adversarial clothes? If so, this method seems to have a great limitation; how to address this?
>
> Answer: (1) We use the same neural SMPL model to estimate the pose and shape parameters for each subject. (2) For the transferability to unseen subjects, Appendix B.8 and Figure 19 show the attack results on unseen subjects. We train the adversarial patch on subject 1 and then test the ASR on subject 2. The attack success rate decreased by 20\%-35\%. We believe this is due to significant differences in body shapes and motion habits between different subjects, which caused the drop in success rate.
> (3) The current method needs finetuning the UV-volume model for each individual based on their specific shape parameters. One way to address this is to sample diverse shape parameters during the optimization process to generate human bodies with a variety of body shapes. By doing this, the adversarial clothing can be trained to generalize across different body shapes. However, this may significantly increase the convergence time and computational complexity of the training process. With the rapid development of dynamic NeRF techniques, we expect the computational time and cost of finetuning a dynamic NeRF will continue decreasing, such as combining UV-Volume with the SHERF[1], making this approach more feasible in the future.
>
> Reference:
> [1]SHERF: Generalizable Human NeRF from a Single Image, ICCV2023

---

> > ### Comment · Reviewer_mhf9 · 2024-11-22
> >
> > Thank the authors for the response.
> > 1. Could you respond to the Weakness 1 as well?
> > 2. How is the finetuned patch different from the pre-finetuned (trained on open source data) with respect to the physical adversarial effectiveness as well as the visual appearance?
> > 3. Is it necessary to optimize the patch on open-source data? What if you directly train with the specific UV-volume model for the target person? Please also refer to Weakness 1 here since the evaluation of w./w.o. the GMM model seems a little problematic.

---

> > > ### Author Response · Authors · 2024-11-25
> > >
> > > Dear Reviewer mhf9，
> > >
> > > Thank you very much for your valuable follow-up comments on our submission! We have clarified and responded to your follow-up questions, and would like to know if our responses satisfy you. We are happy to reply to any unclear parts, if any, since the last day for reviewers to ask questions to authors is approaching!
> > >
> > > Best, Authors

---

> ### Author Response · Authors · 2024-11-23
> **### Comments to Reviewer mhf9**
>
> Thank you for your follow-up comments. We sincerely appreciate the opportunity to clarify these points in greater detail.
>
> **Response to W1**: Thanks for your valuable advice. In Appendix D of the revised version, we evaluate the transferability of our attack using five video sequences from the ZJU-Mocap dataset, which were not seen during the training phase. **Figure 21 compares the ASR of three methods, Random Noises, DAP (Guesmi et al., 2024) and UV-Attack across five pose sequences (#386, #387,#390, #392, #394) from the ZJU-Mocap dataset.** UV-Attack consistently achieves the highest ASR, peaking at 0.91 on #387, while DAP shows moderate performance and Random performs poorly. This highlights the effectiveness of UV-Attack on unseen datasets. **We also visualize the DAP and UV-Attacks in Figure 22.** The first column displays the original pose images. By extracting the pose parameters, we generate human images wearing normal clothing and adversarial clothing. The results demonstrate that our attack achieves a high success rate, with an average ASR of 79% across five unseen pose sequences, even on previously unobserved poses, significantly outperforming the SOTA DAP method, which only achieves an ASR of 38\%. This highlights our approach's strong effectiveness and robustness to unseen poses.
>
> 2. How is the finetuned patch different from the pre-finetuned (trained on open source data) with respect to the physical adversarial effectiveness as well as the visual appearance?
>
> Answer: In Figure 19, we apply the patch trained on open-source data directly to the target person. It is shown that the patch exhibits a certain level of transferability across different individuals. However, due to variations in body shapes, fine-tuning can further improve the patch's success rate by 20%-30% by better adapting it to the target individual. The visual appearance of the finetuned patch is similar to the original adversarial patch.
>
> 3. Is it necessary to optimize the patch on open-source data? What if you directly train with the specific UV-volume model for the target person? Please also refer to Weakness 1 here since the evaluation of w./w.o. the GMM model seems a little problematic.
>
>  Answer: We chose to fine-tune the patch instead of retraining it from scratch primarily **due to considerations of training time**. As shown in Figure 19, the patch exhibits a certain level of transferability across different individuals. However, due to variations in body shapes, fine-tuning can further improve the patch's success rate by better adapting it to the target individual. Training from scratch for 200 epochs requires approximately 2 hours, whereas fine-tuning for just 50 epochs achieves a similarly low loss value in only about 30 minutes.
>
> Reference:
> [1] Amira Guesmi, Ruitian Ding, Muhammad Abdullah Hanif, Ihsen Alouani, and Muhammad Shafique.
> Dap: A dynamic adversarial patch for evading person detectors. In Proceedings of the IEEE/CVF
> Conference on Computer Vision and Pattern Recognition, pp. 24595–24604, 2024

---

> > ### Comment · Reviewer_mhf9 · 2024-11-26
> >
> > Thank the authors for their comprehensive studies of the ablations and their response. I recognize the contribution of this paper to the field and will raise my rating.

---

> > > ### Author Response · Authors · 2024-11-26
> > >
> > > We are very grateful to the reviewers for the appreciation of our work.

---

### Official Review · Reviewer_knkd · 2024-11-04

**Soundness:** 3
**Presentation:** 3
**Contribution:** 3
**Rating:** 6
**Confidence:** 3

**Summary:**

This paper introduces a novel adversarial attack pipeline for person detection, termed UV-Attack. First, it innovatively incorporates a NeRF-based UV mapping method into the person detection adversarial attack pipeline, enabling the generation of diverse human UV maps and textures with varying poses and camera perspectives. Second, it leverages a diffusion model to generate adversarial patches, which are then interpolated with textures produced by 3D human models to render adversarial human images. Finally, comprehensive experiments are conducted across various mainstream detectors and scenarios, benchmarking the proposed method against existing approaches.

**Strengths:**

This paper presents a novel approach to generating adversarial samples for human detection. Firstly, it cleverly utilizes the UV map and texture of the human model to introduce adversarial patches, enabling efficient rendering onto RGB images of the human figure for multi-view attacks. Secondly, it leverages the robust generative priors of diffusion models to perform interpolation at the texture level. Finally, this method achieves significant accuracy improvements in dynamic and multi-pose scenarios by sampling various human poses and perspectives, further validating the effectiveness of the proposed approach.

**Weaknesses:**

This paper has shortcomings in the presentation of experimental details and data, which may lead to confusion regarding the reproducibility and generalizability of the findings.

**Questions:**

1. In your training for a specific detector, you mentioned that you collected 100 different backgrounds from both indoor and outdoor scenarios. Could you provide more details about this data and its sources? In other words, how significantly does this data impact the effectiveness of the adversarial sample generation?
2. Your pipeline requires sampling SMPL pose parameters from a Gaussian Mixture Model (GMM). When modeling the GMM, you need to input the target video and the human pose dataset. What is the purpose of including the target video?
3. When validating the ASR, you average class labels except for "person" in the COCO dataset. During training, is the diffusion model condition kept consistent throughout?

---

> ### Author Response · Authors · 2024-11-21
> **Comments to Reviewer knkd**
>
> ### Comments to Reviewer knkd
> We thank you for your reviews and address your concerns as follows.
>
> Weaknesses:
> This paper has shortcomings in the presentation of experimental details and data, which may lead to confusion regarding the reproducibility and generalizability of the findings.
>
> Answer: Thanks for your advice.  We present more experimental details and results in the Appendix.
> (1) We evaluate our attack on more detectors with different backbones, as shown in A1 of the Appendix of our revised version. These detectors include the ViTDET, RTMDET, Co-DETR-R50 and Co-DETR-SWIN. Moreover, we compare our attack with the latest person detection attacks, Adv-Camou (CVPR2023) and DAP (CVPR2024).
> (2) We do more ablation studies in Appendix B, including the influence of different environments, the sensitivities to different poses, the influence of classifier-free guidance, and the attack results on robust models.
> (3) We present more examples in the physical attacks in Appendix C.
>
> Questions:
> 1. In your training for a specific detector, you mentioned that you collected 100 different backgrounds from both indoor and outdoor scenarios. Could you provide more details about this data and its sources? In other words, how significantly does this data impact the effectiveness of the adversarial sample generation?
>
> Answer: We collected these backgrounds through the Internet. To evaluate the influence of the background images on the adversarial sample generation, we generate the adversarial patches using the same background dataset used by the Adv-Camou (https://github.com/WhoTHU/Adversarial_camou). Then we test the attack results on a different background dataset (Indoor09), which contains 67 Indoor categories, and a total of 15620 images. We present the attack results in Appendix B.1. It is shown that our attack can achieve high ASRs for different indoor scenarios.
>
> 2. Your pipeline requires sampling SMPL pose parameters from a Gaussian Mixture Model (GMM). When modeling the GMM, you need to input the target video and the human pose dataset. What is the purpose of including the target video?
>
> Answer: The reason we include action videos of the target individual is that everyone's walking habits may differ. Although the dataset contains various actions, it may still differ from the walking style of the target individual. To better achieve evasion of detection, we train on the habitual actions of the target individual, which can effectively improve the success rate of evasion in the physical world.
>
> 3. When validating the ASR, you average class labels except for "person" in the COCO dataset. During training, is the diffusion model condition kept consistent throughout?
>
> Answer: During training, the diffusion model condition is kept consistent throughout. Moreover, Appendix B.6 and Figure 17 shows the training loss curves across epochs under different prompts. Although the convergence speed varies slightly among different prompts, they all converge to a relatively small value. This indicates that our method is effective across different prompts.

---

> > ### Author Response · Authors · 2024-11-25
> >
> > Dear Reviewer knkd，
> >
> > Thank you very much for your valuable comments on our submission! We have clarified and responded to your questions, and would like to know if our responses satisfy you. We are happy to reply to any unclear parts, if any, since the last day for reviewers to ask questions to authors is approaching!
> >
> > Best, Authors

---

> ### Comment · Reviewer_knkd · 2024-11-29
>
> Thank you for your detailed response and the additional experiments addressing the concerns raised in my initial review. Your efforts have successfully clarified the issues. I would maintain my positive score.

---

### Official Review · Reviewer_1muW · 2024-11-04

**Soundness:** 3
**Presentation:** 2
**Contribution:** 3
**Rating:** 6
**Confidence:** 2

**Summary:**

The paper presents UV-Attack, a novel physical-world adversarial attack for person detection systems. The core idea is leveraging dynamic Neural Radiance Fields (NeRF) for UV mapping to generate adversarial images across varying human actions and viewpoints. The attack modifies clothing textures through UV maps rather than traditional RGB images, allowing real-time, practical texture edits. A new Expectation over Pose Transformation (EoPT) loss is introduced to improve the attack's success rate for unseen human poses. The paper highlights the potential of using dynamic NeRF and UV mapping for adversarial attacks on non-rigid objects like human bodies.

**Strengths:**

1. The use of dynamic NeRF-based UV mapping for adversarial attacks is an innovative approach, addressing the challenge of human movement.

2. The approach can generate adversarial textures in real-time, making it feasible for real-world attacks

3. The proposed method outperforms previous adversarial attacks, achieving a high ASR across varied poses and detectors, particularly in free-pose settings.

4. The method’s ability to handle diverse human poses through EoPT and UV mapping may enhance its robustness

**Weaknesses:**

1. The method heavily depends on pretrained stable diffusion models for generating adversarial patches, which might limit its generalizability to other model architectures.

2. The attack pipeline involves multiple steps, including dynamic NeRF, UV mapping, and diffusion models, which increases the complexity and may pose practical limitations in some applications.

3. While the paper claims success in physical-world attacks, the physical-world experiments are limited to a few environments and detection models.

4. The paper shows good results on some detectors but does not fully address how transferable the attack is to other models not tested in the experiments. The impact of domain shift (e.g., different datasets) is also not well explored.

**Questions:**

In addition to the weaknesses, please refer to the following:

1. How sensitive is the attack to the specific poses, say sampled from the Gaussian Mixture Model (GMM)? Would other pose distributions significantly affect the results? There is limited statistical insight in the paper.

2.  Does the complexity of clothing textures (e.g., different patterns or colors) impact the effectiveness of the attack?

3. How well does UV-Attack generalize to detection models not tested in the paper, particularly beyond YOLO and FastRCNN variants? Could there be a model-specific bias in the attack’s success rate? This needs to be addressed in detail.

4. There is limited discussion on potential defenses against UV-Attack, which is crucial for the broader adversarial machine learning community. A discussion or experiment on how robust the attack is to adversarial training or other defenses would make the paper more impactful.

---

> ### Author Response · Authors · 2024-11-21
> **Comments to Reviewer 1mUW**
>
> Comments to Reviewer 1mUW:
> Thank you for your suggestions. We have made revisions to address your concerns in the revised version. The revision can be summarized as follows:
> 1. We evaluate our attack on more detectors in Appendix A1.
> 2. We test the ASR and mAP on a new indoor scenario dataset in Appendix B.1 and in different physical environments in Appendix C.
> 3. In Appendix B4, we analyze the relationship between image complexity and ASR.
>
> W1. The generalizability to other model architectures.
> Answer. Our method is generalizable to any latent diffusion model, such as any finetuned diffusion model. Moreover, because the textures of the UV-Volumes are separable from the NeRF model, it can be easily adapted to existing adversarial patch generation methods.
>
> W2: The complexity and practical limitations in some applications.
> Answer: Although our method contains different modules, these modules allow us to solve human detection attacks with high accuracy under different actions for the first time. All of these steps work together to make our approach far superior to previous methods.
>
> W3. While the paper claims success in physical-world attacks, the physical-world experiments are limited to a few environments and detection models.
>
> Answer: **In Appendix B.1, we evaluated the digital ASR and mAP on a new indoor scenario dataset**, which includes 10 different scenarios with over 100 images per scenario, covering diverse settings such as bedrooms, bathrooms, airports, churches, offices, libraries, museums, and more. **In Appendix C, we evaluate the physical attack success rate in three different scenarios.** Furthermore, we tested the attack success rate on additional detection models, and the results are presented in Appendix A1.
>
> W4. The paper shows good results on some detectors but does not fully address how transferable the attack is to other models not tested in the experiments. The impact of domain shift (e.g., different datasets) is also not well explored.
>
> A4. We test the white-box ASR and transferability of our attack on 12 different detectors in the Appendix, including the state-of-the-art detector, Co-DETR. For the impact of domain shift, we evaluate the attack performance on a different background dataset, Indoor-CVPR09, and show the results in Appendix B1.
>
> Q1. How sensitive is the attack to the specific poses, say sampled from the Gaussian Mixture Model (GMM)? Would other pose distributions significantly affect the results? There is limited statistical insight in the paper.
>
> Answer: As shown in Figure 11 in Appendix B.2, we analyze the sensitivity of ASR to different poses sampled from a GMM distribution. The results indicate that a high ASR is consistently achieved across poses corresponding to various components. For other pose distributions, we tested uniform and Gaussian distributions in our experiments. The uniform distribution converges particularly slowly because it generates many unrealistic human poses. The ASR of the Gaussian distribution on the test set is slightly lower than that of the GMM distribution. This is because the poses sampled from the Gaussian distribution lack diversity, while human poses exhibit a multimodal distribution across many dimensions, as shown in Figure 10 in the Appendix.
>
> Q2. Does the complexity of clothing textures (e.g., different patterns or colors) impact the effectiveness of the attack?
>
> Answer: In Appendices B3 and B4, we qualitatively and quantitatively analyze the relationship between image complexity and attack success rate. Specifically, as shown in Figure 14, we find that in the initial few epochs, total variance increases and then stabilizes, while mAP continuously decreases. This suggests that our method can achieve a high ASR with lower total variance compared to perturbation-based methods.
>
> Q3. How well does UV-Attack generalize to detection models not tested in the paper, particularly beyond YOLO and FastRCNN variants? Could there be a model-specific bias in the attack’s success rate? This needs to be addressed in detail.
>
> Answer: As mentioned above, we test our attack on state-of-the-art detectors in the Appendix of the revised paper, including ViTDET and Co-DETR-SWIN which use transformers as backbones. To avoid the model-specific bias in ASR, we evaluate the attack effectiveness on two metrics, ASR and mAP, as shown in Figure 7 and Figure 8 of our paper.
>
> Q4. There is limited discussion on potential defenses against UV-Attack, which is crucial for the broader adversarial machine learning community. A discussion or experiment on how robust the attack is to adversarial training or other defenses would make the paper more impactful.
>
> Answer: Thanks for your advice. To evaluate the robustness of the attack to adversarail training, we test the transferability on a robust detector RobustDet. As shown in Appendix B.5 and Figure 15, although RobustDet is resistant to PGD and DAG attacks, its mAP drops from 1.0 to 0.85 on our adversarial examples.

---

> > ### Comment · Reviewer_1muW · 2024-11-21
> > **Post-rebuttal**
> >
> > Thanks for the responses. Since my concerns are addressed by the authors, specifically as they added new experiments, which illustrate the effectiveness of the method more, I increase my rating by 1 point.

---

> > > ### Author Response · Authors · 2024-11-22
> > >
> > > We sincerely thank you for your thoughtful feedback and for taking the time to review our responses and additional experiments. We are glad to hear that our clarifications and the new experiments addressed your concerns.

---

### Meta-Review · Area_Chair_SdJr · 2024-12-16

**Metareview:**

This paper proposes a novel adversarial attack for person detection systems, leveraging dynamic NeRF and UV mapping to generate realistic and adaptable adversarial textures. By manipulating clothing textures through UV maps, the method allows real-time texture edits across various human poses and viewpoints, achieving high attack success rates. The key novelties include an Expectation over Pose Transformation (EoPT) loss and the use of diffusion models for texture generation, which outperform existing methods in both digital and physical-world scenarios. The reviewers highlighted the extensive and thorough experimentation across various person detection models and environments, including both white-box and black-box scenarios, as well as digital and physical settings. The use of dynamic NeRF-based UV mapping for adversarial attacks is also noted as an innovative approach, effectively addressing the challenge of human movement with adaptable and realistic texture modifications. Additionally, the clarity of the writing and the real-time feasibility of generating adversarial textures for practical attacks are appreciated by reviewers. In summary, all reviewers are positive about this work.

**Additional Comments On Reviewer Discussion:**

The reviewers raised several concerns, including the limited scope of physical-world experiments, the heavy reliance on pretrained stable diffusion models, the absence of a null model for comparison, unclear training details, and the setup of the target model. They also highlighted that the paper focuses on attack success rates without addressing visual stealth, noting that UV-Attack’s textures may appear unnatural and easily detectable in real-world settings. Additionally, discrepancies were observed between the reported success rates and those found in the original publications. The rebuttal effectively addressed all of these issues, with all reviewers recommending "accept" or "marginally above the acceptance threshold."

---

### Decision · Program_Chairs · 2025-01-22

Accept (Poster)